# On Dynamic Programming Decompositions of Static Risk Measures in Markov Decision Processes

**Jia Lin Hau**
University of New Hampshire
Durham, NH
jialin.hau@unh.edu

**Erick Delage**
HEC Montréal
Montréal (Québec)
erick.delage@hec.ca

**Mohammad Ghavamzadeh**[*]
Amazon
Palo Alto, CA
ghavamza@amazon.com

**Marek Petrik**
University of New Hampshire
Durham, NH
mpetrik@cs.unh.edu

## Abstract

Optimizing static risk-averse objectives in Markov decision processes is difficult because they do not admit standard dynamic programming equations common in Reinforcement Learning (RL) algorithms. Dynamic programming decompositions that augment the state space with discrete risk levels have recently gained popularity in the RL community. Prior work has shown that these decompositions are optimal when the risk level is discretized sufficiently. However, we show that these popular decompositions for Conditional-Value-at-Risk (CVaR) and Entropic-Value-at-Risk (EVaR) are inherently suboptimal regardless of the discretization level. In particular, we show that a saddle point property assumed to hold in prior literature may be violated. However, a decomposition does hold for Value-at-Risk and our proof demonstrates how this risk measure differs from CVaR and EVaR. Our findings are significant because risk-averse algorithms are used in high-stakes environments, making their correctness much more critical.

## 1 Introduction

Risk-averse reinforcement learning (RL) seeks to provide a risk-averse policy for high-stakes real-world decision problems. These high-stake domains include autonomous driving (Jin et al., 2019; Sharma et al., 2020), robot collision avoidance (Ahmadi et al., 2021; Hakobyan and Yang, 2021), liver transplant timing (Köse, 2016), HIV treatment (Keramati et al., 2020; Zhong, 2020), unmanned aerial vehicle (UAV) (Choudhry et al., 2021), and investment liquidation (Min et al., 2022), to name a few. Because these domains call for reliable solutions, risk-averse algorithms must be based on solid theoretical foundations. This is one reason why monetary risk measures, such as Value-at-Risk (VaR) and Conditional Value-at-Risk (CVaR), have become pervasive in risk-averse RL (Prashanth and Fu, 2022). Indeed, risk measures such as CVaR are known to be coherent (Artzner et al., 1999) with respect to a set of fundamental axioms that define how risk should be quantified and have been adopted as gold standards in banking regulations (Basel Committee on Banking Supervision, 2019).

Introducing risk-averse objectives in Markov decision processes (MDPs)—the primary model used in RL—is challenging. Dynamic programming, the linchpin of most RL algorithms, cannot be used directly to optimize a risk measure like VaR or CVaR in MDPs. One line of work tackles this challenge by exploiting the primal representation of risk measures and augmenting the state space

---

[*]The work was done prior to joining Amazon, while the author was at Google Research.

37th Conference on Neural Information Processing Systems (NeurIPS 2023).

of their dynamic programs (DPs) with an additional parameter that typically represents the total cumulative reward up to the current point (Bäuerle and Ott, 2011; Boda et al., 2004; Chow and Ghavamzadeh, 2014; Filar et al., 1995; Hau et al., 2023; Lin et al., 2003; Wu and Lin, 1999; Xu and Mannor, 2011). Even when the original MDP is finite, this DP requires computing the value function for a continuous state space, and thus, has been considered inefficient in practice (Chapman et al., 2022; Chow et al., 2015; Li et al., 2022).

Another line of recent work leverages the dual representation to produce a *risk-level decomposition* of risk measures (Pflug and Pichler, 2016). Using this decomposition, numerous authors have derived DPs for common risk measures and integrated them within various RL algorithms (Chapman et al., 2019, 2022; Chow et al., 2015; Ding and Feinberg, 2022, 2023; Ni and Lai, 2022; Rigter et al., 2021; Stanko and Macek, 2019). Although this risk-level decomposition requires augmenting the state space with a continuous parameter, this parameter is naturally bounded between 0 and 1. It has been generally accepted, with several *tentative* proofs supporting this claim (Chow et al., 2015; Li et al., 2022), that these DPs recover the optimal policy if we can discretize the augmented state space sufficiently finely. Moreover, it is believed that one can use the optimal value function from this DP to recover the policies that are optimal for the full range of risk levels.

In this paper, we make a surprising discovery that numerous claims of optimality of risk-level decompositions published in the past several years are incorrect. Even when one discretizes the augmented state space arbitrarily finely, most risk-level DPs are not guaranteed to recover the optimal value function and policy. There are several reasons why existing arguments fail. As the most common reason, several papers assume that a certain saddle point property holds, either explicitly (Chow et al., 2015) or implicitly (Ding and Feinberg, 2022, 2023). We show that this property does not generally hold, invalidating the optimality of DPs, as hinted at in Chapman et al. (2019, 2022). This finding directly refutes the claimed or hypothesized optimality of algorithms proposed in many recent research papers and pre-prints, such as Chapman et al. (2019); Chow et al. (2015); Ding and Feinberg (2022, 2023); Rigter et al. (2021); Stanko and Macek (2019). Our results also affect applications of these algorithms, such as automated vehicle motion planning (Jin et al., 2019). We also identify gaps in related decompositions (Li et al., 2022; Ni and Lai, 2022) and propose how to fix them.

We make the following contributions in this paper. *First*, we show in Section 3 that the popular DP for optimizing CVaR in MDPs may not recover the optimal value function and policy regardless of how finely one discretizes the risk level in the augmented states. This method was first proposed in Chow et al. (2015) but adopted widely afterwards (Chapman et al., 2019; Ding and Feinberg, 2022, 2023; Rigter et al., 2021; Stanko and Macek, 2019). The simple counterexample in this section contradicts the optimality claims in Chow et al. (2015); Ding and Feinberg (2022). We hypothesize that prior work missed this issue because the CVaR DP works for policy evaluation and only fails when one uses it to optimize policies based on the "risk-to-go value function". Therefore, our results do not contradict the original decomposition in Pflug and Pichler (2016) that only applies to policy evaluation. We give a new independent and simple proof that the CVaR decomposition indeed works when evaluating a fixed policy.

*Second*, we show in Section 4 that the DP for optimizing the Entropic-Value-at-Risk in MDPs, proposed by Ni and Lai (2022), does not compute the correct value function even when the policy is fixed. Although EVaR has not been as popular as CVaR, it has been gaining attention in recent years (Hau et al., 2023). We give an example that contradicts the correctness claims of the risk-level decomposition for EVaR in Ni and Lai (2022). The gap that we identify with this objective applies to both policy evaluation and policy optimization. Furthermore, we prove a new, correct EVaR decomposition for policy evaluation. Unfortunately, the EVaR decomposition fails and is sub-optimal when applied to policy optimization, similar to CVaR.

*Third*, we propose an *optimal dynamic program* for policy optimization of VaR in Section 5. Our DP is based on a risk-level decomposition that closely resembles the quantile MDP decomposition in Li et al. (2022) but corrects for several technical inaccuracies. The derivation shows why VaR stands apart from coherent risk measures like CVaR and EVaR. VaR is unique in that the decomposition can be constructed directly from the primal formulation of the risk measure, which avoids the complications that arise in the robust formulations used in CVaR and EVaR decompositions.

It is important to note that the correctness of DPs that augment the state space with the accumulated rewards is unaffected by our results (Bäuerle and Ott, 2011; Chow and Ghavamzadeh, 2014; Chow et al., 2018; Hau et al., 2023). These DPs use the *primal* risk measure representation and do not suffer

from the same saddle point issue as the augmentation methods that use the *dual* representation of the risk measures, such as the one in Chow et al. (2015).

## 2   Preliminaries

This section summarizes relevant properties of monetary risk measures and outlines how they are typically used in the context of solving MDPs.

**Monetary Risk Measures**   We restrict our attention to probability spaces with a finite outcome space $\Omega$ such that $|\Omega| = m$ for some $m \in \mathbb{N}$. We use $\mathbb{X} = \mathbb{R}^m$ to denote the space of real-valued random variables. To improve the clarity of probabilistic claims, we always adorn random variables with a tilde, such as $\tilde{x} \in \mathbb{X}$. In finite spaces, we can represent any random variable $\tilde{x} \in \mathbb{X}$ as a vector $\boldsymbol{x} \in \mathbb{R}^m$. We also use $\boldsymbol{q} \in \Delta_m$ to represent a probability distribution over $\Omega$ where $\Delta_m$ represents the $m$-dimensional probability simplex. Using this notation, we can write that $\mathbb{E}[\tilde{x}] = \boldsymbol{q}^\top \boldsymbol{x}$.

A *monetary risk measure* $\psi \colon \mathbb{X} \to \mathbb{R}$ assigns a real value to each real-valued random variable in a way that it is monotone and cash-invariant (Follmer and Schied, 2016; Shapiro et al., 2014). A risk measure can be seen as a generalization of the expectation operator $\mathbb{E}[\cdot]$ that also takes into account the uncertainty in the random variable. In this work, we define all risk measures for random variables $\tilde{x}$ that represent *rewards*. Thus, the risk-averse decision-maker aims to choose actions that maximize the value of the risk measure, i.e., a higher value of risk measure represents a lower exposure to risk.

We consider three monetary risk measures common in RL. Perhaps the most well-known measure is *Value-at-Risk* (VaR), which is defined for a risk-level $\alpha \in [0, 1]$ and a random variable $\tilde{x} \in \mathbb{X}$ in modern literature as (e.g., Follmer and Schied 2016; Shapiro et al. 2014)

$$\text{VaR}_\alpha [\tilde{x}] = \sup\ \{z \in \mathbb{R} \mid \mathbb{P}[\tilde{x} < z] \le \alpha\} = \inf\ \{z \in \mathbb{R} \mid \mathbb{P}[\tilde{x} \le z] > \alpha\}. \tag{1}$$

Note that $\text{VaR}_1[\tilde{x}] = \infty$. The equality between the two definitions holds, for example, by Follmer and Schied (2016, remark A.20).

Another popular risk measure is the *Conditional-value-at-Risk* (CVaR), which is defined for a risk level $\alpha \in [0, 1]$ and a random variable $\tilde{x} \in \mathbb{X}$ distributed as $\tilde{x} \sim \boldsymbol{q}$ as (e.g., Follmer and Schied 2016, definition 11.8 and Shapiro et al. 2014, eq. 6.23)

$$\text{CVaR}_\alpha [\tilde{x}] = \sup_{z \in \mathbb{R}}\ \left(z - \alpha^{-1}\mathbb{E}\left[z - \tilde{x}\right]_+\right) = \inf\ \left\{\boldsymbol{\xi}^\top \boldsymbol{x} \mid \boldsymbol{\xi} \in \Delta_m, \alpha \cdot \boldsymbol{\xi} \le \boldsymbol{q}\right\}, \tag{2}$$

with $\text{CVaR}_0[\tilde{x}] = \operatorname{ess\,inf}[\tilde{x}]$ and $\text{CVaR}_1[\tilde{x}] = \mathbb{E}[\tilde{x}]$. The equality above follows from standard conjugacy arguments for finite probability spaces (Follmer and Schied, 2016). Note that our CVaR definition applies to $\tilde{x}$ that represents rewards and assumes that a higher value of the risk measure is preferable to a lower value. Other CVaR formulations exist in the literature, but they induce identical preferences for appropriately chosen rewards and risk level $\alpha$.

Finally, the *entropic value at risk* (EVaR), with $\text{EVaR}_0[\tilde{x}] = \operatorname{ess\,inf}[\tilde{x}]$ and $\text{EVaR}_1[\tilde{x}] = \mathbb{E}[\tilde{x}]$, is defined for $\alpha \in (0, 1]$ as (Ahmadi-Javid, 2012)

$$\begin{aligned}
\text{EVaR}_\alpha [\tilde{x}] &= \sup_{\beta > 0}\ -\frac{1}{\beta} \log\left(\alpha^{-1}\mathbb{E}\left[\exp\left(-\beta \cdot \tilde{x}\right)\right]\right) \\
&= \inf\ \left\{\boldsymbol{\xi}^T \boldsymbol{x} \mid \boldsymbol{\xi} \in \Delta_m, \boldsymbol{\xi} \ll \boldsymbol{q}, \text{KL}(\boldsymbol{\xi} \| \boldsymbol{q}) \le -\log \alpha\right\},
\end{aligned} \tag{3}$$

where KL is the standard KL-divergence defined for each $\boldsymbol{x}, \boldsymbol{y} \in \Delta_m$ as $\text{KL}(\boldsymbol{x} \| \boldsymbol{y}) = \sum_{\omega \in \Omega} x_\omega \log\left(x_\omega / y_\omega\right)$. This definition is valid only when $\boldsymbol{x}$ is absolutely continuous with respect to $\boldsymbol{y}$, which is denoted as $\boldsymbol{x} \ll \boldsymbol{y}$ and corresponds to $y_\omega = 0 \Rightarrow x_\omega = 0$ for each $\omega \in \Omega$.

**Risk Averse MDPs**   A Markov decision process (MDP) is a sequential decision model that underlies most of RL (Puterman, 2005). We consider finite MDPs with states $\mathcal{S} = \{s_1, \dots, s_S\}$ and actions $\mathcal{A} = \{a_1, \dots, a_A\}$. After taking an action in a state, the agent transitions to the next state according to a transition probability function $p \colon \mathcal{S} \times \mathcal{A} \to \Delta_\mathcal{S}$ such that $p(s, a, s')$ represents the transition probability from $s \in \mathcal{S}$ to $s' \in \mathcal{S}$ after taking $a \in \mathcal{A}$. We use $\boldsymbol{p}_{s,a} = p(s, a, \cdot) \in \Delta_\mathcal{S}$ to denote the vector of transition probabilities. The initial state $\tilde{s}_0$ is distributed according to $\hat{\boldsymbol{p}} \in \Delta_S$. To avoid divisions by 0 that are not central to our claims, we assume that $\hat{p}_s > 0$ for each $s \in \mathcal{S}$. Finally,

the reward function is $r \colon \mathcal{S} \times \mathcal{A} \times \mathcal{S} \to \mathbb{R}$, where $r(s, a, s')$ represents the deterministic reward associated with the transition to $s'$ from $s$ after taking an action $a$.

The most general solution to an MDP is a *history-dependent randomized* policy $\pi$ which maps a sequence of observed states and actions $s^0, a^0, s^1, a^1, \ldots, s^t$ to a distribution over the next action $a^t$. It is well-known that with risk-neutral objectives, there always exists an optimal stationary—depends only on the last state—deterministic policy (Puterman, 2005). When the objective is risk-averse, like VaR, or CVaR, there may not exist an optimal stationary or deterministic policy. Hence, we use the symbol $\Pi$ to denote the set of history-dependent randomized policies in the remainder of the paper.

This paper focuses on the *finite-horizon objective* in which the agent aims to compute policies that optimize the sum of rewards over a known horizon $T$. We further restrict our attention to the objective with horizon $T = 1$. It turns out that having a single time step is sufficient to derive our counterexamples to existing dynamic programs. Moreover, deriving the decompositions with $T = 1$ makes it possible to avoid technicalities caused by history-dependent policies, which could distract us from the main ideas presented in this work. Our results an be extended to general horizons $T > 1$ and the discounted infinite-horizon objectives using standard techniques (Chow et al., 2015).

With horizon $T = 1$, the set of randomized history-dependent policies is $\Pi = \{\pi \colon \mathcal{S} \to \Delta_A\}$. The symbol $\pi(s, a)$ denotes the probability of action $a$ in a state $s$, and $\pi(s) = \pi(s, \cdot) \in \Delta_A$ denotes the $A$-dimensional vector of action probabilities in a state $s$. Given a risk measure $\psi$ with a risk level $\alpha \in [0, 1]$, the finite-horizon risk-averse value of a policy $\pi \in \Pi$ is computed as

$$v_0^\pi(\alpha) = \psi_\alpha^{\tilde{a} \sim \pi(\tilde{s})} \left[ r(\tilde{s}, \tilde{a}, \tilde{s}') \right], \tag{4}$$

where the superscript in $\psi_\alpha^{\tilde{a} \sim \pi(\tilde{s})}$ specifies the distribution of the random action. Throughout the paper, we generally use $\tilde{s}$ to denote the random state at time $t = 0$ and $\tilde{s}'$ to denote the random state at time $t = 1$. In risk-neutral objectives, when $\psi = \mathbb{E}$, one can use the tower property of the expectation operator and define a value function $v_t$ for each time step $t$ (Puterman, 2005), but this property does not hold in most static risk measures (Hau et al., 2023). The term *policy evaluation* in the remainder of the paper refers to computing the value in (4).

The goal in an MDP is to compute an *optimal* value function and a policy that attains it. In risk-averse MDPs, this goal is formalized as the following risk-averse optimization

$$v_0^\star(\alpha) = \max_{\pi \in \Pi} v_0^\pi(\alpha) = \max_{\pi \in \Pi} \psi_\alpha^{\tilde{a} \sim \pi(\tilde{s})} \left[ r(\tilde{s}, \tilde{a}, \tilde{s}') \right], \tag{5}$$

with the *optimal policy* $\pi^\star$ being any policy that attains the maximum in (5). As with policy evaluation, when $\psi = \mathbb{E}$, the optimal value function $v_t^\star$ can be defined for each time-step $t$ (Puterman, 2005), but this is impossible in general for common risk measures, like VaR and CVaR. The term *policy optimization* in the remainder of the paper refers to computing the value and the maximizer in (5).

In the remainder of the paper, we study dynamic programming algorithms proposed to solve the policy evaluation problem in (4) and policy optimization problem in (5). In general, these algorithms build on risk-level decomposition (Pflug and Pichler, 2016) of risk measures to define a value function $v_t^\pi(s, \alpha)$ for each time step $t \in [T]$, state $s \in \mathcal{S}$, and risk-level $\alpha \in [0, 1]$ (Chow et al., 2015). The value function represents the risk-adjusted sum of rewards that can be obtained if starting in a state $s \in \mathcal{S}$ at time $t$ and a risk level $\alpha$. For example, one would define the value function as $v_1^\pi(s, \alpha) = \psi_\alpha^{\tilde{a} \sim \pi(s)}[r(s, \tilde{a}, \tilde{s}')]$ and compute $v_0^\pi$ using a Bellman operator $T_\alpha^\pi$ as $v_0^\pi(\alpha) = (T^\pi v_1^\pi)(\alpha)$. In risk-neutral objectives, the Bellman operator is defined as $(T^\pi(v_1^\pi))(\alpha) = \mathbb{E}[v_1^\pi(\tilde{s}, \alpha)]$, with $\alpha \in \{1\}$, but in risk-averse formulations the operator definition is more complex. The remainder of the paper discusses the decompositions and the operator for CVaR, EVaR, and VaR risk measures respectively.

## 3 CVaR: Decomposition Fails in Policy Optimization

In this section, we show that a common CVaR decomposition proposed in Chow et al. (2015) and used to optimize risk-averse policies is inherently sub-optimal regardless of how closely one discretizes the state space. The following proposition represents one of the key results used to decompose the risk measure in multi-stage decision-making.

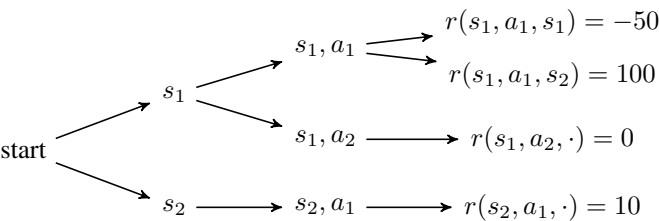

Figure 1: Rewards of MDP $M_C$ used in the proof of Theorem 3.2. The dot indicates that the rewards are independent of the next state.

**Proposition 3.1** (lemma 22 in Pflug and Pichler 2016). *Suppose that $\pi \in \Pi$ and $\tilde{s} \sim \hat{p}$, $\tilde{a} \sim \pi(\tilde{s})$, $\tilde{s}' \sim p_{s,a}$. Then,*

$$\mathrm{CVaR}_\alpha\left[r(\tilde{s}, \tilde{a}, \tilde{s}')\right] \quad = \quad \min_{\zeta \in \mathcal{Z}_C} \sum_{s \in \mathcal{S}} \zeta_s \, \mathrm{CVaR}_{\alpha \zeta_s \hat{p}_s^{-1}}\left[r(s, \tilde{a}, \tilde{s}') \mid \tilde{s} = s\right], \tag{6}$$

*where the state $s$ on the right-hand side is not random and*

$$\mathcal{Z}_C \;=\; \{\zeta \in \Delta_S \mid \alpha \cdot \zeta \leq \hat{p}\}. \tag{7}$$

The notation in Proposition 3.1 differs superficially from lemma 22 in Pflug and Pichler (2016). Specifically, our CVaR is defined for rewards rather than costs, the meaning of our $\alpha$ corresponds to $1 - \alpha$ in Pflug and Pichler (2016), and we use $\xi_s = z_s \hat{p}_s$ as the optimization variable. We include a simple proof of Proposition 3.1 for completeness in Appendix A.1.

The decomposition in Proposition 3.1 is important because it shows that the CVaR evaluation can be formulated as a dynamic program. The theorem shows that CVaR at time $t = 0$ decomposes into a convex combination of CVaR values at $t = 1$. Recursively repeating this process, one can formulate a dynamic program for any finite time horizon $T$. Because the risk level at $t = 1$ differs from the level at $t = 0$ and depends on the optimal $\zeta$, one must compute CVaR values for all (or many) risk-levels $\alpha \in [0, 1]$ at time $t = 1$. As a result, the dynamic program includes an additional state variable that represents the current risk level.

Chow et al. (2015) proposed to adapt the decomposition in Proposition 3.1 to policy optimization as

$$\max_{\pi \in \Pi} \; \mathrm{CVaR}_\alpha^{\tilde{a} \sim \pi(\tilde{s})}[r(\tilde{s}, \tilde{a}, \tilde{s}')] \;=\; \max_{\pi \in \Pi} \min_{\zeta \in \mathcal{Z}_C} \sum_{s \in \mathcal{S}} \zeta_s \left(\mathrm{CVaR}_{\alpha \zeta_s \hat{p}_s^{-1}}^{\tilde{a} \sim \pi(s)}[r(s, \tilde{a}, \tilde{s}') \mid \tilde{s} = s]\right)$$

$$\overset{??}{=} \; \min_{\zeta \in \mathcal{Z}_C} \sum_{s \in \mathcal{S}} \zeta_s \left(\max_{d \in \Delta_\mathcal{A}} \mathrm{CVaR}_{\alpha \zeta_s \hat{p}_s^{-1}}^{\tilde{a} \sim d}[r(s, \tilde{a}, \tilde{s}') \mid \tilde{s} = s]\right). \tag{8}$$

They used the decomposition in (8) to formulate a dynamic program with the current risk level as an additional state variable. We prove in the following theorem that the second equality in (8) marked with question marks is false in general.

**Theorem 3.2.** *There exists an MDP and a risk level $\alpha \in [0, 1]$ such that*

$$\max_{\pi \in \Pi} \; \mathrm{CVaR}_\alpha^{\tilde{a} \sim \pi(\tilde{s})}[r(\tilde{s}, \tilde{a}, \tilde{s}')] \;<\; \min_{\zeta \in \mathcal{Z}_C} \sum_{s \in \mathcal{S}} \zeta_s \left(\max_{d \in \Delta_\mathcal{A}} \mathrm{CVaR}_{\alpha \zeta_s \hat{p}_s^{-1}}^{\tilde{a} \sim d}[r(s, \tilde{a}, \tilde{s}') \mid \tilde{s} = s]\right). \tag{9}$$

Before proving Theorem 3.2, we discuss its implications. First, Theorem 3.2 contradicts theorems 5 and 7 in Chow et al. (2015) and shows that their algorithm is inherently sub-optimal regardless of the resolution of the discretization. Theorem 3.2 also contradicts the optimality of the accelerated dynamic program proposed in Stanko and Macek (2019). The result of (Chow et al., 2015) was exploited as is in Chapman et al. (2019), Ding and Feinberg (2022), and Jin et al. (2019) to propose DP reductions, and extended, without proof, in (Rigter et al., 2021) to the context of a Bayesian MDP.

Finally, it is important to emphasize that Theorem 3.2 only applies to the policy optimization setting and does not contradict Proposition 3.1, which holds for the evaluation of policies that assign the same action distribution to each history of states and actions (i.e., policies that are independent of the hypothesized values of $\zeta$).

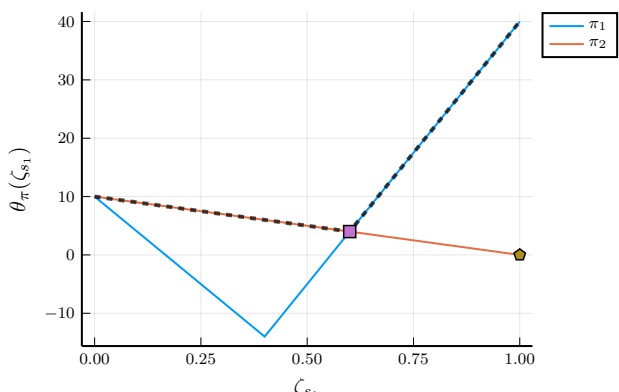

Figure 2: The functions $\theta_{\pi_1}(\cdot)$ and $\theta_{\pi_2}(\cdot)$ used in the CVaR counterexample in the proof of Theorem 3.2. The dashed line shows the function $\zeta_{s_1} \mapsto \max_{\pi \in \{\pi_1, \pi_2\}} \theta_\pi([\zeta_{s_1}, 1 - \zeta_{s_1}])$.

*Proof.* Let $\alpha = 0.5$ and consider the MDP $M_\mathrm{C}$ in Figure 1. In state $s_1$, both actions $a_1$ and $a_2$ are available, and in state $s_2$, only action $a_1$ is available. The MDP's rewards are

$$
\begin{aligned}
r(s_1, a_1, s_1) &= -50, & r(s_1, a_1, s_2) &= 100, \\
r(s_1, a_2, s_1) &= r(s_1, a_2, s_2) = 0, & r(s_2, a_1, s_1) &= r(s_2, a_1, s_2) = 10 \,.
\end{aligned}
$$

The transition probabilities in $M_\mathrm{C}$ are

$$
p(s_1, a_1, s_1) = 0.4, \qquad\qquad p(s_1, a_1, s_2) = 0.6 \,,
$$

and the initial distribution is uniform: $\hat{p}_{s_1} = \hat{p}_{s_2} = 0.5$.

To simplify the notation, we define $\theta_\pi \colon \mathcal{Z}_\mathrm{C} \to \mathbb{R}$ for each $\pi \in \Pi$ and $\boldsymbol\zeta \in \mathcal{Z}_\mathrm{C}$ as

$$
\theta_\pi(\boldsymbol\zeta) \;=\; \sum_{s \in \mathcal{S}} \zeta_s \, \mathrm{CVaR}_{\alpha \zeta_s \hat{p}_s^{-1}}^{\tilde{a} \sim \boldsymbol\pi(s)} [r(s, \tilde{a}, \tilde{s}') \mid \tilde{s} = s] \,.
$$

Because CVaR is convex in the distribution (Delage et al., 2019) and any distribution for $r(\tilde{s}, \tilde{a}, \tilde{s}')$ obtained from a policy $\pi \in \Pi$ is a mixture of the distributions of $r(\tilde{s}, a_1, \tilde{s}')$ and $r(\tilde{s}, a_2, \tilde{s}')$, it is sufficient to consider only *deterministic* policies (there exists an optimal deterministic policy). Thus, we can reformulate the *left-hand side* of (9) in terms of $\theta_\pi(\boldsymbol\zeta)$ as

$$
\begin{aligned}
\max_{\boldsymbol\pi \in \Pi} \mathrm{CVaR}_\alpha^{\tilde{a} \sim \boldsymbol\pi(\tilde{s})} [r(\tilde{s}, \tilde{a}, \tilde{s}')] &= \max_{\pi \in \{\pi_1, \pi_2\}} \mathrm{CVaR}_\alpha^{\tilde{a} \sim \boldsymbol\pi(\tilde{s})} [r(\tilde{s}, \tilde{a}, \tilde{s}')] \\
&= \max_{\pi \in \{\pi_1, \pi_2\}} \min_{\boldsymbol\zeta \in \mathcal{Z}_\mathrm{C}} \sum_{s \in \mathcal{S}} \zeta_s \cdot \mathrm{CVaR}_{\alpha \zeta_s \hat{p}_s^{-1}}^{\tilde{a} \sim \boldsymbol\pi(s)} [r(s, \tilde{a}, \tilde{s}') \mid \tilde{s} = s] \\
&= \max_{\pi \in \{\pi_1, \pi_2\}} \min_{\boldsymbol\zeta \in \mathcal{Z}_\mathrm{C}} \theta_\pi(\boldsymbol\zeta) \,,
\end{aligned}
$$

with $\pi_1(s, a_1) = 1 - \pi_1(s, a_2) = 1$ and $\pi_2(s, a_2) = 1 - \pi_2(s, a_1) = 1$, for all $s \in \mathcal{S}$. The functions $\theta_{\pi_1}(\cdot)$ and $\theta_{\pi_2}(\cdot)$ are depicted in Figure 2. Similarly, the *right-hand side* of (9) can be expressed using the convexity of CVaR in the distribution by algebraic manipulation as

$$
\min_{\boldsymbol\zeta \in \mathcal{Z}_\mathrm{C}} \sum_{s \in \mathcal{S}} \zeta_s \max_{\boldsymbol{d} \in \Delta_A} \mathrm{CVaR}_{\alpha \zeta_s \hat{p}_s^{-1}}^{\tilde{a} \sim \boldsymbol{d}} [r(s, \tilde{a}, \tilde{s}') \mid \tilde{s} = s] = \min_{\boldsymbol\zeta \in \mathcal{Z}_\mathrm{C}} \max_{\pi \in \{\pi_1, \pi_2\}} \theta_\pi(\boldsymbol\zeta) \,.
$$

Using the notation introduced above and the sufficiency of optimizing over deterministic policies only, the inequality in (9) becomes

$$
\max_{\pi \in \{\pi_1, \pi_2\}} \min_{\boldsymbol\zeta \in \mathcal{Z}_\mathrm{C}} \theta_\pi(\boldsymbol\zeta) \;<\; \min_{\boldsymbol\zeta \in \mathcal{Z}_\mathrm{C}} \max_{\pi \in \{\pi_1, \pi_2\}} \theta_\pi(\boldsymbol\zeta) \,. \tag{10}
$$

Figure 2 demonstrates the inequality in (10) numerically, with the rectangle representing the left-hand side maximum and the pentagon representing the right-hand side minimum. The dashed line represents the function $\boldsymbol\zeta \mapsto \max_{\pi \in \{\pi_1, \pi_2\}} \theta_\pi(\boldsymbol\zeta)$.

To show the strict inequality in (10) formally, we evaluate the functions $\theta_{\pi_1}(\cdot)$ and $\theta_{\pi_2}(\cdot)$ for MDP $M_C$. The function $\theta_{\pi_2}(\cdot)$ is linear because the CVaR applies to a constant, and CVaR is translation invariant. The function $\theta_{\pi_1}(\cdot)$ is piecewise-linear and convex, and its slope can be computed using the subgradient that for each $s \in \mathcal{S}$ and $\hat{\boldsymbol{\zeta}} \in \mathcal{Z}_C$ satisfies (Chow et al., 2015)

$$\partial_{\zeta_s} \hat{\zeta}_s \, \mathrm{CVaR}_{\alpha \hat{p}_s^{-1} \hat{\zeta}_s} \left[ r(s, \tilde{a}, \tilde{s}') \mid \tilde{s} = s \right] \ni \mathrm{VaR}_{\alpha \hat{p}_s^{-1} \hat{\zeta}_s} \left[ r(s, \tilde{a}, \tilde{s}') \mid \tilde{s} = s \right] .$$

Simple algebraic manipulation then shows that

$$\theta_{\pi_1}(\boldsymbol{\zeta}) = \max \left\{ 10 - 60 \, \zeta_{s_1}, \ 90 \, \zeta_{s_1} - 50 \right\}, \qquad \theta_{\pi_2}(\boldsymbol{\zeta}) = 10 - 10 \, \zeta_{s_1},$$

and $\mathcal{Z}_C = \Delta_{\mathcal{S}}$, which implies that $\zeta_{s_1} \in (0,1)$. Therefore, by algebraic manipulation, we get the desired strict inequality

$$0 = \max_{\pi \in \{\pi_1, \pi_2\}} \min_{\boldsymbol{\zeta} \in \mathcal{Z}_C} \theta_\pi(\boldsymbol{\zeta}) < \min_{\boldsymbol{\zeta} \in \mathcal{Z}_C} \max_{\pi \in \{\pi_1, \pi_2\}} \theta_\pi(\boldsymbol{\zeta}) = 4,$$

where 0 and 4 are represented by the pentagon and rectangle in Figure 2, respectively. □

In summary, the decomposition in Proposition 3.1 cannot be exploited in policy optimization because the inequality in the derivation above may not be tight:

$$\max_{\pi \in \Pi} \mathrm{CVaR}_\alpha^{\tilde{a} \sim \boldsymbol{\pi}(\tilde{s})} [r(\tilde{s}, \tilde{a}, \tilde{s}') \mid \tilde{s} = s] = \max_{\pi \in \Pi} \min_{\boldsymbol{\zeta} \in \mathcal{Z}_C} \sum_{s \in \mathcal{S}} \zeta_s \, \mathrm{CVaR}_{\alpha \zeta_s \hat{p}_s^{-1}}^{\tilde{a} \sim \boldsymbol{\pi}(\tilde{s})} [r(s, \tilde{a}, \tilde{s}') \mid \tilde{s} = s]$$

$$\leq \min_{\boldsymbol{\zeta} \in \mathcal{Z}_C} \max_{\pi \in \Pi} \sum_{s \in \mathcal{S}} \zeta_s \, \mathrm{CVaR}_{\alpha \zeta_s \hat{p}_s^{-1}}^{\tilde{a} \sim \boldsymbol{\pi}(\tilde{s})} [r(s, \tilde{a}, \tilde{s}') \mid \tilde{s} = s]$$

$$= \min_{\boldsymbol{\zeta} \in \mathcal{Z}_C} \sum_{s \in \mathcal{S}} \zeta_s \max_{\boldsymbol{d} \in \Delta_A} \mathrm{CVaR}_{\alpha \zeta_s \hat{p}_s^{-1}}^{\tilde{a} \sim \boldsymbol{d}} [r(s, \tilde{a}, \tilde{s}') \mid \tilde{s} = s],$$

where the last equality follows from the interchangeability property of optimization and expected value (Shapiro et al., 2014, theorem 7.92).

Finally, we omit to comment on the validity of the CVaR decomposition in Li et al. (2022) given that it considers a different measure than the CVaR defined in Equation (1). Namely, their measure takes the form:

$$\widetilde{\mathrm{CVaR}}_\alpha[\tilde{x}] = \inf_{z \in \mathbb{R}} \left( z + (1-\alpha)^{-1} \mathbb{E} \left[ \tilde{x} - z \right]_+ \right) = - \mathrm{CVaR}_{1-\alpha} \left[ -\tilde{x} \right],$$

which is not a coherent risk measure.

## 4 EVaR: Decomposition Fails for Policy Evaluation

In this section. we show that a decomposition for EVaR proposed in Ni and Lai (2022) is inexact even when considering the policy evaluation setting. Ni and Lai (2022) recently proposed a decomposition of EVaR for a fixed $\pi \in \Pi$ with $\tilde{a} \sim \boldsymbol{\pi}(\tilde{s})$ and a risk level $\alpha \in (0,1]$ as

$$\mathrm{EVaR}_\alpha \left[ r(\tilde{s}, \tilde{a}, \tilde{s}') \right] \overset{??}{=} \min_{\boldsymbol{\xi} \in \mathcal{Z}_E} \sum_{s \in \mathcal{S}} \xi_s \, \mathrm{EVaR}_{\alpha \xi_s \hat{p}_s^{-1}} \left[ r(s, \tilde{a}, \tilde{s}') \mid \tilde{s} = s \right], \tag{11}$$

where $\tilde{s} \sim \hat{\boldsymbol{p}}$, $\tilde{s}' \sim p(\tilde{s}, \tilde{a}, \cdot)$, and

$$\mathcal{Z}_E = \left\{ \boldsymbol{\xi} \in \Delta_{\mathcal{S}} \mid \sum_{s \in \mathcal{S}} \xi_s \log(\xi_s / \hat{p}_s) \leq -\log \alpha, \quad \overbrace{\alpha \cdot \boldsymbol{\xi} \leq \hat{\boldsymbol{p}}}^{\text{implicit in Ni and Lai (2022)}} \right\}. \tag{12}$$

Note that we use variables $\xi_s = z_s \hat{p}_s$ in comparison with $z_s$ in Ni and Lai (2022).

The constraint $\alpha \cdot \boldsymbol{\xi} \leq \hat{\boldsymbol{p}}$ in (12) was not stated explicitly in Ni and Lai (2022) but is necessary because $\mathrm{EVaR}_{\alpha'} [\cdot]$ is defined only for $\alpha' \in [0,1]$. When $\alpha' = \alpha \xi_s \hat{p}_s^{-1}$ in (11) it must also satisfy for each $s \in \mathcal{S}$ that

$$\alpha' \leq 1 \quad \Leftrightarrow \quad \alpha \xi_s \hat{p}_s^{-1} \leq 1 \quad \Leftrightarrow \quad \alpha \cdot \xi_s \leq \hat{p}_s.$$

This additional constraint on $\boldsymbol{\xi}$ implies that $\mathcal{Z}_E \subseteq \mathcal{Z}_C$, for the $\mathcal{Z}_C$ defined in (7).

We claim in the following theorem (see Appendix A.2 for the proof) that the equality in (11) does not hold even in the policy evaluation setting.

**Theorem 4.1.** *There exists an MDP with a single action and $\alpha \in (0,1]$ such that*

$$\mathrm{EVaR}_\alpha\left[r(\tilde{s}, a_1, \tilde{s}')\right] \quad < \quad \min_{\boldsymbol{\xi} \in \mathcal{Z}_{\mathrm{E}}} \sum_{s \in \mathcal{S}} \xi_s \, \mathrm{EVaR}_{\alpha \xi_s \hat{p}_s^{-1}}\left[r(s, a_1, \tilde{s}') \mid \tilde{s} = s\right], \tag{13}$$

*the set $\mathcal{Z}_{\mathrm{E}}$ defined by (12).*

Theorem 4.1 demonstrates a stronger failure mode than the one in Theorem 3.2 (for CVaR policy optimization) since it applies to both policy evaluation and policy optimization settings.

We propose a correct decomposition of EVaR in the following theorem and employ it to establish that the decomposition in (11) overestimates the actual value of EVaR (see Appendix A.3 for a proof).

**Theorem 4.2.** *Given any finite MDP with horizon $T = 1$ and $\alpha \in (0,1]$, we have that*

$$\mathrm{EVaR}_\alpha\left[r(\tilde{s}, \tilde{a}, \tilde{s}')\right] \quad = \quad \inf_{\boldsymbol{\zeta} \in (0,1]^S, \boldsymbol{\xi} \in \mathcal{Z}'_{\mathrm{E}}(\boldsymbol{\zeta})} \sum_{s \in \mathcal{S}} \xi_s \, \mathrm{EVaR}_{\zeta_s}\left[r(s, \tilde{a}, \tilde{s}') \mid \tilde{s} = s\right],$$

*where*

$$\mathcal{Z}'_{\mathrm{E}}(\boldsymbol{\zeta}) = \left\{\boldsymbol{\xi} \in \Delta_S \mid \boldsymbol{\xi} \ll \hat{\boldsymbol{p}}, \sum_{s \in \mathcal{S}} \xi_s \left(\log(\xi_s/\hat{p}_s) - \log(\zeta_s)\right) \leq -\log \alpha\right\}.$$

*Moreover, EVaR can be upper-bounded as*

$$\mathrm{EVaR}_\alpha\left[r(\tilde{s}, \tilde{a}, \tilde{s}')\right] \quad \leq \quad \min_{\boldsymbol{\xi} \in \mathcal{Z}_{\mathrm{E}}} \sum_{s \in \mathcal{S}} \xi_s \, \mathrm{EVaR}_{\alpha \xi_s \hat{p}_s^{-1}}\left[r(s, \tilde{a}, \tilde{s}') \mid \tilde{s} = s\right]. \tag{14}$$

## 5 VaR: Decomposition Holds for Policy Evaluation and Optimization

In this section, we discuss a dynamic program decomposition for VaR whose decomposition resembles those for CVaR and EVaR described in Sections 3 and 4. We provide a new proof of the VaR decomposition to elucidate the differences that make it optimal in contrast to CVaR and EVaR decompositions. Our VaR decomposition closely resembles the quantile MDP approach in Li et al. (2022) with a few technical modifications that can significantly impact the computed value.

To contrast the typical definition of VaR with the quantile definition in Li et al. (2022), it is helpful to summarize how VaR is related to the quantile of a random variable. Let $\mathfrak{q} \in \mathbb{R}$ define as the *$\alpha$-quantile* of $\tilde{x} \in \mathbb{X}$ when

$$\mathbb{P}\left[\tilde{x} \leq \mathfrak{q}\right] \geq \alpha \quad \text{and} \quad \mathbb{P}\left[\tilde{x} < \mathfrak{q}\right] \leq \alpha. \tag{15}$$

In general, the set of quantiles is an interval $[\mathfrak{q}_{\tilde{x}}^-(\alpha), \mathfrak{q}_{\tilde{x}}^+(\alpha)]$ with the bounds computed as (Follmer and Schied, 2016, appendix A.3)

$$\mathfrak{q}_{\tilde{x}}^-(\alpha) = \sup\left\{z \mid \mathbb{P}\left[\tilde{x} < z\right] < \alpha\right\} = \inf\left\{z \mid \mathbb{P}\left[\tilde{x} \leq z\right] \geq \alpha\right\},$$
$$\mathfrak{q}_{\tilde{x}}^+(\alpha) = \inf\left\{z \mid \mathbb{P}\left[\tilde{x} \leq z\right] > \alpha\right\} = \sup\left\{z \mid \mathbb{P}\left[\tilde{x} < z\right] \leq \alpha\right\}.$$

Note that when the distribution of $\tilde{x}$ is absolutely continuous (atomless), then $\mathfrak{q}_{\tilde{x}}^+(\alpha) = \mathfrak{q}_{\tilde{x}}^-(\alpha)$ and the quantile is unique. The following example illustrates a simple setting in which the quantile is not unique.

*Example* 1 (Bernoulli random variable). Consider a Bernoulli random variable $\tilde{e}$ such that $\tilde{e} = 1$ and $\tilde{e} = 0$ with equal (50%) probabilities. Then, any value $q \in [0,1]$ is a valid 0.5-quantile because

$$\mathfrak{q}_{\tilde{e}}^-(0.5) = \inf_{z \in \mathbb{R}}\ \left\{z \mid \mathbb{P}\left[\tilde{e} \leq z\right] \geq 0.5\right\} = \inf_{z \in \mathbb{R}}\ \left\{z \mid z \geq 0\right\} = 0$$
$$\mathfrak{q}_{\tilde{e}}^+(0.5) = \sup_{z \in \mathbb{R}}\ \left\{z \mid \mathbb{P}\left[\tilde{e} \geq z\right] \geq 0.5\right\} = \sup_{z \in \mathbb{R}}\ \left\{z \mid z \leq 1\right\} = 1.$$

The objective in Li et al. (2022) is to maximize the quantile operator $Q_\alpha \colon \mathbb{X} \to \mathbb{R}$ defined for a reward random variable $\tilde{x} \in \mathbb{X}$ and a risk level $\alpha \in [0,1]$ as

$$Q_\alpha(\tilde{x}) = \inf_{z \in \mathbb{R}}\ \left\{z \mid \mathbb{P}\left[\tilde{x} \leq z\right] \geq \alpha\right\}. \tag{16}$$

The quantile operator $Q_\alpha$ and VaR differ in which quantile of the random variable they consider:

$$Q_\alpha(\tilde{x}) = \mathfrak{q}_{\tilde{x}}^-(\alpha), \qquad \text{but} \qquad \mathrm{VaR}_\alpha\left[\tilde{x}\right] = \mathfrak{q}_{\tilde{x}}^+(\alpha). \tag{17}$$

As a result, the quantile MDP objective in (16) coincides with the VaR value only when the quantile is unique, which is not always the case, as shown in Example 1.

**Theorem 5.1.** *Let $\tilde{y}\colon \Omega \to [N]$ be a random variable distributed as $\hat{\boldsymbol{p}} = (\hat{p}_i)_{i=1}^{N}$ with $\hat{p}_i > 0$. Then for any random variable $\tilde{x} \in \mathbb{X}$, we have*

$$\mathrm{VaR}_\alpha\left[\tilde{x}\right] \quad = \quad \sup_{\boldsymbol{\zeta} \in \Delta_N} \left\{ \min_i \ \mathrm{VaR}_{\alpha \zeta_i \hat{p}_i^{-1}}\left[\tilde{x} \mid \tilde{y} = i\right] \mid \alpha \cdot \boldsymbol{\zeta} \le \hat{\boldsymbol{p}} \right\}, \tag{18}$$

*where we interpret the minimum to evaluate to $\infty$ if all terms are infinite, which only occurs if $\alpha = 1$.*

*Proof.* We decompose VaR using the definition in (1) as

$$\mathrm{VaR}_\alpha\left[\tilde{x}\right] = \sup \ \{z \in \mathbb{R} \mid \mathbb{P}\left[\tilde{x} < z\right] \le \alpha\} \overset{(a)}{=} \sup \left\{ z \in \mathbb{R} \mid \sum_{i=1}^{N} \mathbb{P}\left[\tilde{x} < z \mid \tilde{y} = i\right] \cdot \hat{p}_i \le \alpha \right\}$$

$$\overset{(b)}{=} \sup \left\{ z \in \mathbb{R} \mid \boldsymbol{\zeta} \in [0,1]^N, \ \mathbb{P}\left[\tilde{x} < z \mid \tilde{y} = i\right] \le \zeta_i, \forall i \in [N], \ \sum_{i=1}^{N} \zeta_i \hat{p}_i \le \alpha \right\}$$

$$\overset{(c)}{=} \sup \left\{ \sup \ \{z \in \mathbb{R} \mid \mathbb{P}\left[\tilde{x} < z \mid \tilde{y} = i\right] \le \zeta_i, \ \forall i \in [N]\} \mid \boldsymbol{\zeta} \in [0,1]^N, \ \sum_{i=1}^{N} \zeta_i \hat{p}_i \le \alpha \right\}$$

$$= \sup \left\{ \sup \ \bigcap_{i \in [N]} \{z \in \mathbb{R} \mid \mathbb{P}\left[\tilde{x} < z \mid \tilde{y} = i\right] \le \zeta_i\} \mid \boldsymbol{\zeta} \in [0,1]^N, \ \sum_{i=1}^{N} \zeta_i \hat{p}_i \le \alpha \right\}$$

$$\overset{(d)}{=} \sup \left\{ \min_{i \in [N]} \sup \ \{z \in \mathbb{R} \mid \mathbb{P}\left[\tilde{x} < z \mid \tilde{y} = i\right] \le \zeta_i\} \mid \boldsymbol{\zeta} \in [0,1]^N, \ \sum_{i=1}^{N} \zeta_i \hat{p}_i \le \alpha \right\}$$

$$\overset{(e)}{=} \sup \left\{ \min_{i \in [N]} \ \mathrm{VaR}_{\zeta_i}\left[\tilde{x} \mid \tilde{y} = i\right] \mid \boldsymbol{\zeta} \in [0,1]^N, \ \sum_{i=1}^{N} \zeta_i \hat{p}_i \le \alpha \right\}.$$

We decompose the probability $\mathbb{P}\left[\tilde{x} < z\right]$ as a marginal of the conditional probabilities $\mathbb{P}\left[\tilde{x} < z \mid \tilde{y} = i\right]$ and $\hat{p}_i = \mathbb{P}\left[\tilde{y} = i\right]$ in step (a) and then lower-bound them by an auxiliary variable $\zeta_i$ in step (b). In step (c) we replace the joint supremum over $z$ and $\boldsymbol{\zeta}$ by sequential suprema, and then we replace the supremum of an intersection by the minimum of the suprema of sets in (d). The equality in (d) holds because $\mathbb{P}\left[\tilde{x} < z\right]$ is monotone and, therefore, the sets $\{z \in \mathbb{R} \mid \mathbb{P}\left[\tilde{x} < z \mid \tilde{y} = i\right] \le \zeta_i\}$ are nested. Finally, step (e) follows from the definition of VaR in (1). $\square$

Focusing on the finite MDP with horizon $T = 1$, we can show that the decomposition proposed in Theorem 5.1 is amenable to policy optimization. The main difference between the VaR decomposition and CVaR is that the former VaR was expressed as a supremum instead of an infimum over quantile levels $\boldsymbol{\zeta}$. For VaR, changing the order of maximum ($\pi$) and supremum ($\boldsymbol{\zeta}$) does not suffer from a potential gap, but changing the order of maximum ($\pi$) and infimum/minimum ($\boldsymbol{\zeta}$) in CVaR does suffer from such a gap as shown in Theorem 3.2.

The following theorem (proved in Appendix A.4) summarizes the decomposition for VaR.

**Theorem 5.2.** *Given any finite MDP with horizon $T = 1$ and $\alpha \in [0,1]$, we have*

$$\max_{\pi \in \Pi} \ \mathrm{VaR}_\alpha^{\tilde{a} \sim \boldsymbol{\pi}(\tilde{s})}[r(\tilde{s}, \tilde{a}, \tilde{s}')] = \sup_{\boldsymbol{\zeta} \in \Delta_S} \left\{ \min_{s \in \mathcal{S}} \max_{\boldsymbol{d} \in \Delta_A} \mathrm{VaR}_{\alpha \zeta_s \hat{p}_s^{-1}}^{\tilde{a} \sim \boldsymbol{d}}\left[r(s, \tilde{a}, \tilde{s}') \mid \tilde{s} = s\right] \mid \alpha \cdot \boldsymbol{\zeta} \le \hat{\boldsymbol{p}} \right\}.$$

For completeness, Appendix C extends Theorem 5.2 to a setting with horizon $T > 1$, providing dynamic programming equations and a definition of the optimal policy. These equations are analogous to the equations presented in (Li et al., 2022) yet we obtain them using the accurate definition and decomposition of $\mathrm{VaR}_\alpha$.

We finally present below a valid decomposition for the lower quantile MDP, used officially as the objective in (Li et al., 2022) (see Appendix A.5 for a proof).

**Proposition 5.3.** *Given any finite MDP with horizon $T = 1$ and some $\alpha \in [0,1]$, we have that:*

$$\max_{\pi \in \Pi} Q_\alpha^{\tilde{a} \sim \boldsymbol{\pi}(\tilde{s})}(r(\tilde{s}, \tilde{a}, \tilde{s}')) = \sup_{\boldsymbol{\zeta} \in [0,1]^S} \left\{ \min_{s \in \mathcal{S}: \zeta_s < 1} \max_{\boldsymbol{d} \in \Delta_A} Q_{\zeta_s}^{\tilde{a} \sim \boldsymbol{d}}(r(s, \tilde{a}, \tilde{s}') \mid \tilde{s} = s) \mid \sum_{s=1}^{S} \zeta_s \hat{p}_s < \alpha \right\}.$$

We note that the difference with the result presented in (Li et al., 2022) resides in the constraint imposed on $\zeta$ that replaces the weak inequality with a strict one. In fact, this strict versus weak inequality is the main distinguishing factor between the decompositions for the lower and upper quantiles.

# 6 Conclusion

This paper shows that a popular decomposition approach to solving MDPs with CVaR and EVaR objectives is suboptimal despite the claims to the contrary. This suboptimality arises from a saddle-point gap when *optimizing policy*. We also prove that a similar decomposition approach is optimal for policy optimization and evaluation when solving MDPs with the VaR objective. The decomposition is optimal because VaR does not involve the same saddle point problem as CVaR and EVaR.

Our findings are significant because practitioners who make risk-averse decisions in high-stakes scenarios need to have confidence in the correctness of the algorithms they use. Our work raises awareness that popular static CVaR and EVaR MDP algorithms are suboptimal, and their analyses are inaccurate. We hope the results we present in our paper will increase the scrutiny of dynamic programming methods for risk-averse MDPs and motivate research into alternative approaches, such as the parametric dynamic programs.

### Acknowledgments

We thank Yinlam Chow for his insightful comments that inspired this paper's topic and results. The work in the paper was supported, in part, by NSF grants 2144601 and 2218063. In addition, this work was performed in part while Marek Petrik was a visiting researcher at Google Research. Finally, E. Delage acknowledges the support from the Canadian Natural Sciences and Engineering Research Council and the Canada Research Chair program [Grant RGPIN-2022-05261 and CRC-2018-00105].

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

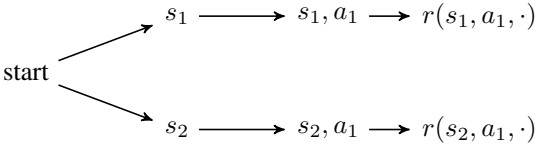

Figure 3: Rewards of the MDP $M_E$ used in the proof of Theorem 4.1. The dot indicates that the rewards are independent of the next state.

## A Proofs

### A.1 Proof of Proposition 3.1

Suppose that $\alpha > 0$; the decomposition for $\alpha = 0$ holds readily because $\mathrm{CVaR}_0 [\tilde{x}] = \operatorname{ess\,inf}[\tilde{x}]$.

To streamline the notation, Define a random variable $\tilde{x} = r(\tilde{s}, \tilde{a}, \tilde{s}')$ over a random space $\Omega = \mathcal{S} \times \mathcal{A} \times \mathcal{S}$ with a probability distribution $\boldsymbol{q} \in \Delta_m$ such that $q_{s,a,s'} = \hat{p}_s \cdot \pi(s,a) \cdot p(s,a,s')$. The value $\boldsymbol{x}$ is the vector representation of the random variable $\tilde{x}$ and $\boldsymbol{\xi}_s = \boldsymbol{\xi}_{s,\cdot,\cdot} \in \mathbb{R}^{S \cdot A}$ for $\boldsymbol{\xi} \in \mathbb{R}^m$ is a vector that corresponds to the subset of the elements of $\Omega$ in which the first element is some $s \in \mathcal{S}$. The vectors $\boldsymbol{x}_s = \boldsymbol{x}_{s,\cdot,\cdot} \in \mathbb{R}^{S \cdot A}$ and $\boldsymbol{q}_s = \boldsymbol{q}_{s,\cdot,\cdot} \in \mathbb{R}^{S \cdot A}$ are defined analogously to $\boldsymbol{\xi}_s$.

Starting with the CVaR definition in (2) and introducing an auxiliary variable $\zeta$ we get that

$$\mathrm{CVaR}_\alpha [\tilde{x}] = \min_{\boldsymbol{\xi} \in \Delta_m} \left\{ \boldsymbol{x}^\top \boldsymbol{\xi} \mid \alpha \boldsymbol{\xi} \leq \boldsymbol{q} \right\} = \min_{\boldsymbol{\xi} \in \Delta_m, \boldsymbol{\zeta} \in \mathbb{R}^S} \left\{ \boldsymbol{x}^\top \boldsymbol{\xi} \mid \alpha \boldsymbol{\xi} \leq \boldsymbol{q}, \, \zeta_s = \mathbf{1}^\top \boldsymbol{\xi}_s, \forall s \in \mathcal{S} \right\}$$

$$= \min_{\boldsymbol{\xi} \in \Delta_m, \boldsymbol{\zeta} \in \Delta_S} \left\{ \boldsymbol{x}^\top \boldsymbol{\xi} \mid \alpha \boldsymbol{\xi} \leq \boldsymbol{q}, \, \zeta_s = \mathbf{1}^\top \boldsymbol{\xi}_s, \, \alpha \zeta_s \leq \hat{p}_s, \, \forall s \in \mathcal{S} \right\}$$

$$= \min_{\boldsymbol{\xi} \in \mathbb{R}_+^\Omega, \boldsymbol{\zeta} \in \mathcal{Z}_C} \left\{ \boldsymbol{x}^\top \boldsymbol{\xi} \mid \alpha \boldsymbol{\xi} \leq \boldsymbol{q}, \, \zeta_s = \mathbf{1}^\top \boldsymbol{\xi}_s, \, \forall s \in \mathcal{S} \right\} .$$

In the derivation above, we replaced the infimum by a minimum because $\Omega$ is finite, introduced a new variable $\zeta$, derived implied constraints on $\zeta$, and then dropped superfluous constraints on $\boldsymbol{\xi}$. Continuing with the derivation above and noticing that the constraints on each $\boldsymbol{\xi}_s$ are independent given $\zeta$, we get that

$$\mathrm{CVaR}_\alpha [\tilde{x}] = \min_{\boldsymbol{\xi} \in \mathbb{R}_+^\Omega, \boldsymbol{\zeta} \in \mathcal{Z}_C} \left\{ \sum_{s \in \mathcal{S}} \boldsymbol{x}_s^\top \boldsymbol{\xi}_s \mid \alpha \boldsymbol{\xi} \leq \boldsymbol{q}, \, \zeta_s = \mathbf{1}^\top \boldsymbol{\xi}_s, \, \forall s \in \mathcal{S} \right\}$$

$$\overset{(a)}{=} \min_{\boldsymbol{\zeta} \in \mathcal{Z}_C} \sum_{s \in \mathcal{S}} \inf_{\boldsymbol{\xi}_s \in \mathbb{R}_+^{\Omega_s}} \left\{ \boldsymbol{x}_s^\top \boldsymbol{\xi}_s \mid \alpha \boldsymbol{\xi}_s \leq \boldsymbol{q}_s, \, \zeta_s = \mathbf{1}^\top \boldsymbol{\xi}_s \right\}$$

$$\overset{(b)}{=} \min_{\boldsymbol{\zeta} \in \mathcal{Z}_C} \sum_{s \in \mathcal{S}} \zeta_s \cdot \min_{\boldsymbol{\chi} \in \Delta_{S \cdot A}} \left\{ \boldsymbol{x}_s^\top \boldsymbol{\chi} \mid \alpha \hat{p}_s^{-1} \zeta_s \chi_{a,s'} \leq \hat{p}_s^{-1} q_{s,a,s'}, \, \forall a \in \mathcal{A}, s' \in \mathcal{S} \right\}$$

$$\overset{(c)}{=} \min_{\boldsymbol{\zeta} \in \mathcal{Z}_C} \sum_{s \in \mathcal{S}} \zeta_s \cdot \mathrm{CVaR}_{\alpha \zeta_s \hat{p}_s^{-1}} [\tilde{x} \mid \tilde{s} = s] .$$

The step (a) follows from the interchangeability principle (Shapiro et al., 2014, theorem 7.92), and the step (b) follows by substituting $\xi_{s,a,s'} = \zeta_s \chi_{a,s'}$ taking care when $\zeta_s = 0$ and multiplying both sides of the inequality by $\hat{p}_s^{-1} > 0$. Finally, in step (c), the random variable $\tilde{x} = r(\tilde{s}, \tilde{a}, \tilde{s}')$ conditional on $\tilde{s} = s$ is distributed according to $q_{s,a,s'} \hat{p}_s^{-1}$ and the equality follows from the definition of CVaR in (2). $\qquad \square$

### A.2 Proof of Theorem Theorem 4.1

Consider an MDP $M_E$ depicted in Figure 3 with $\mathcal{S} = \{s_1, s_2\}$ and $\mathcal{A} = \{a_1\}$ and a reward function $r(s_1, a_1, \cdot) = 1$ and $r(s_2, a_1, \cdot) = 0$. We abbreviate the rewards to $r(s_1)$ and $r(s_2)$ because they only depend on the originating state. The initial distribution is $\hat{p}_{s_1} = \hat{p}_{s_2} = 0.5$. We finally let $\alpha = 0.75$.

Because $\mathcal{Z}_{\mathrm{E}} \subseteq \mathcal{Z}_{\mathrm{C}}$, the right-hand side of (13) can be lower-bounded by CVaR as

$$
\begin{aligned}
\min_{\boldsymbol{\xi} \in \mathcal{Z}_{\mathrm{E}}} \sum_{s \in \mathcal{S}} \xi_s \, \mathrm{EVaR}_{\alpha \xi_s \hat{p}_s^{-1}} \left[ r(s, a_1, \tilde{s}') \right] &= \min_{\boldsymbol{\xi} \in \mathcal{Z}_{\mathrm{E}}} \sum_{s \in \mathcal{S}} \xi_s r(s) \\
&\geq \min_{\boldsymbol{\xi} \in \mathcal{Z}_{\mathrm{C}}} \sum_{s \in \mathcal{S}} \xi_s r(s) = \mathrm{CVaR}_\alpha \left[ r(\tilde{s}, a_1, \tilde{s}') \right] .
\end{aligned}
\tag{19}
$$

The first equality holds from the positive homogeneity and cash invariance properties of EVaR, and the last equality follows from the dual representation of CVaR (Follmer and Schied, 2016).

Because $\mathrm{EVaR}_\alpha [\tilde{x}] \leq \mathrm{CVaR}_\alpha [\tilde{x}]$ for each $\alpha \in [0,1]$ and $\tilde{x} \in \mathbb{X}$ (see (Ahmadi-Javid, 2012, proposition 3.2)), we can further lower-bound (19) as

$$
\mathrm{EVaR}_\alpha \left[ r(\tilde{s}, a_1, \tilde{s}') \right] \leq \mathrm{CVaR}_\alpha \left[ r(\tilde{s}, a_1, \tilde{s}') \right] \leq \min_{\boldsymbol{\xi} \in \mathcal{Z}_{\mathrm{E}}} \sum_{s \in \mathcal{S}} \xi_s \, \mathrm{EVaR}_{\alpha \xi_s \hat{p}_s^{-1}} \left[ r(s, a_1, \tilde{s}') \right] . \tag{20}
$$

Therefore, (13) holds with an inequality.

To prove by contradiction that the inequality in (13) is strict, suppose that

$$
\mathrm{EVaR}_\alpha \left[ r(\tilde{s}, a_1, \tilde{s}') \right] = \min_{\boldsymbol{\xi} \in \mathcal{Z}_{\mathrm{E}}} \sum_{s \in \mathcal{S}} \xi_s \, \mathrm{EVaR}_{\alpha \xi_s \hat{p}_s^{-1}} \left[ r(s, a_1, \tilde{s}') \right] . \tag{21}
$$

Equalities (21) and (20) imply that $\mathrm{EVaR}_\alpha \left[ r(\tilde{s}, a_1, \tilde{s}') \right] = \mathrm{CVaR}_\alpha \left[ r(\tilde{s}, a_1, \tilde{s}') \right]$ which is false in general (Ahmadi-Javid, 2012).

We now show that EVaR does not equal CVaR even for the categorical distribution of $\tilde{s}$. The CVaR of the return in $M_{\mathrm{E}}$ reduces from (2) to

$$
\mathrm{CVaR}_\alpha \left[ r(\tilde{s}, a_1, \tilde{s}') \right] = \min_{\boldsymbol{\xi} \in \mathcal{Z}_{\mathrm{C}}} \sum_{s \in \mathcal{S}} \xi_s r(s) = \max \left\{ 0, \frac{\hat{p}_{s_1} + \alpha - 1}{\alpha} \right\} . \tag{22}
$$

Since $1 - \alpha = 0.25 < 0.5 = \hat{p}_{s_1}$, then the optimal $\boldsymbol{\xi}^\star$ in (22) is

$$
\boldsymbol{\xi}^\star = \begin{pmatrix} \frac{\hat{p}_{s_1} + \alpha - 1}{\alpha} \\ \frac{1 - \hat{p}_{s_1}}{\alpha} \end{pmatrix} .
$$

Since $\mathrm{KL}(\boldsymbol{\xi}^\star \| \hat{\boldsymbol{p}}) < -\log \alpha$. we have that $\boldsymbol{\xi}^\star$ is in the relative interior of the EVaR feasible region in (3), and, therefore, there exists an $\epsilon > 0$ such that

$$
\mathrm{EVaR}_\alpha \left[ r(\tilde{s}, a_1, \tilde{s}') \right] = \mathrm{CVaR}_\alpha \left[ r(\tilde{s}, a_1, \tilde{s}') \right] - \epsilon < \mathrm{CVaR}_\alpha \left[ r(\tilde{s}, a_1, \tilde{s}') \right] ,
$$

which proves the desired inequality. $\qquad\square$

## A.3  Proof of Corollary 4.2

We start by proposing a new decomposition for EVaR.

**Proposition A.1.** *Given a random variable $\tilde{x} \in \mathbb{X}$ and a discrete variable $\tilde{y} \colon \Omega \to \mathcal{N} = \{1, \ldots, N\}$, with probabilities denoted as $\{\hat{p}_i\}_{i=1}^N$, for any $\alpha \in (0, 1]$ we have that*

$$
\mathrm{EVaR}_\alpha [\tilde{x}] = \inf_{\boldsymbol{\zeta} \in (0,1]^N} \min_{\boldsymbol{\xi} \in \mathcal{Z}_{\mathrm{E}}'(\boldsymbol{\zeta})} \sum_i \xi_i \, \mathrm{EVaR}_{\zeta_i} \left[ \tilde{x} \mid \tilde{y} = i \right] ,
$$

*where*

$$
\mathcal{Z}_{\mathrm{E}}'(\boldsymbol{\zeta}) = \left\{ \boldsymbol{\xi} \in \Delta_N \mid \boldsymbol{\xi} \ll \hat{\boldsymbol{p}}, \sum_{i=1}^N \xi_i (\log(\xi_i / \hat{p}_i) - \log(\zeta_i)) \leq -\log \alpha \right\} .
$$

*Proof.* Let $\boldsymbol{q}$ denote the joint probability distribution of $\tilde{x}$ and $\tilde{y}$. The proof exploits the chain rule of relative entropy (e.g., Cover and Thomas (2006, theorem 2.5.3)), which states that for any probability distributions $\boldsymbol{\eta}, \boldsymbol{q} \in \Delta_\Omega$ with $\boldsymbol{\eta} \ll \boldsymbol{q}$ that

$$
\mathrm{KL}(\boldsymbol{\eta} \| \boldsymbol{q}) = \mathrm{KL}(\boldsymbol{\eta}(\tilde{y}) \| \boldsymbol{q}(\tilde{y})) + \mathrm{KL}(\boldsymbol{\eta}(\tilde{x} | \tilde{y}) \| \boldsymbol{q}(\tilde{x} | \tilde{y})), \tag{23}
$$

where the conditional relative entropy is defined as

$$\mathrm{KL}(\boldsymbol{\eta}(\tilde{x}|\tilde{y})\|\boldsymbol{q}(\tilde{x}|\tilde{y})) = \mathbb{E}^{\boldsymbol{\eta}}\left[\log\frac{\boldsymbol{\eta}(\tilde{x}|\tilde{y})}{\boldsymbol{q}(\tilde{x}|\tilde{y})}\right].$$

with $\mathbb{E}^{\boldsymbol{\eta}}[f(\tilde{x},\tilde{y})]$ as a shorthand notation to indicate that $(\tilde{x},\tilde{y})\sim\boldsymbol{\eta}$. We can now decompose EVaR from its definition in (3) as

$$\mathrm{EVaR}_\alpha\left[\tilde{x}\right] = \inf_{\boldsymbol{\eta}\in\Delta_{m\cdot N}:\boldsymbol{\eta}\ll\boldsymbol{q}}\left\{\mathbb{E}^{\boldsymbol{\eta}}[\tilde{x}]\mid\mathrm{KL}(\boldsymbol{\eta}\mid\boldsymbol{q})\leq-\log\alpha\right\}$$

$$\overset{(a)}{=} \inf_{\boldsymbol{\eta}\in\Delta_{m\cdot N}:\boldsymbol{\eta}\ll\boldsymbol{q}}\left\{\mathbb{E}^{\boldsymbol{\eta}}[\tilde{x}]\mid\mathrm{KL}(\boldsymbol{\eta}(\tilde{y})\|\boldsymbol{q}(\tilde{y}))+\mathrm{KL}(\boldsymbol{\eta}(\tilde{x}|\tilde{y})\|\boldsymbol{q}(\tilde{x}|\tilde{y}))\leq-\log\alpha\right\}$$

$$= \inf_{\boldsymbol{\eta}\in\Delta_{m\cdot N}:\boldsymbol{\eta}\ll\boldsymbol{q}}\left\{\mathbb{E}^{\boldsymbol{\eta}}[\tilde{x}]\mid\mathrm{KL}(\boldsymbol{\eta}(\tilde{y})\|\boldsymbol{q}(\tilde{y}))+\mathbb{E}^{\boldsymbol{\eta}}\left[\mathbb{E}^{\boldsymbol{\eta}}\left[\log\frac{\boldsymbol{\eta}(\tilde{x}|\tilde{y})}{\boldsymbol{q}(\tilde{x}|\tilde{y})}\right]\mid\tilde{y}\right]\leq-\log\alpha\right\}$$

$$\overset{(b)}{=} \inf_{\boldsymbol{\eta}\in\Delta_{m\cdot N},\boldsymbol{\zeta}\in(0,1]^N:\boldsymbol{\eta}\ll\boldsymbol{q}}\left\{\mathbb{E}^{\boldsymbol{\eta}}[\mathbb{E}^{\boldsymbol{\eta}}[\tilde{x}\mid\tilde{y}]]\mid\begin{array}{l}\mathrm{KL}(\boldsymbol{\eta}(\tilde{y})\|\boldsymbol{q}(\tilde{y}))+\mathbb{E}^{\boldsymbol{\eta}}[-\log(\zeta_{\tilde{y}})]\leq-\log\alpha\\\mathbb{P}_{\boldsymbol{\eta}}\left[\mathbb{E}^{\boldsymbol{\eta}}\left[\log(\boldsymbol{\eta}(\tilde{x}|\tilde{y})/\boldsymbol{q}(\tilde{x}|\tilde{y}))\mid\tilde{y}\right]\leq-\log(\zeta_{\tilde{y}})\right]=1\end{array}\right\}$$

$$\overset{(c)}{=} \inf_{\boldsymbol{\xi}\in\Delta_N,\boldsymbol{\zeta}\in(0,1]^N:\boldsymbol{\xi}\ll\hat{\boldsymbol{p}}}\left\{\mathbb{E}^{\boldsymbol{\xi}}[\mathrm{EVaR}_{\zeta_{\tilde{y}}}[\tilde{x}|\tilde{y}]]\mid\mathrm{KL}(\boldsymbol{\xi}\|\hat{\boldsymbol{p}})+\mathbb{E}^{\boldsymbol{\xi}}[-\log(\zeta_{\tilde{y}})]\leq-\log\alpha\right\}$$

$$= \inf_{\boldsymbol{\xi}\in\Delta_N,\boldsymbol{\zeta}\in(0,1]^N:\boldsymbol{\xi}\ll\hat{\boldsymbol{p}}}\left\{\sum_i\xi_i\,\mathrm{EVaR}_{\zeta_i}[\tilde{x}\mid\tilde{y}=i]\mid\sum_{i=1}^N\xi_i\log(\xi_i/\hat{p}_i)-\sum_{i=1}^N\xi_i\log(\zeta_i)\leq-\log\alpha\right\}.$$

Here, we decompose the relative entropy of $\boldsymbol{\eta}$ and $\boldsymbol{q}$ using (23) in step (a) and then use the tower property of the expectation operator in the next step. In step (b), we introduce a variable $\zeta_i$ for each realization of $\tilde{y}=i$ with $i\in\mathcal{N}$ to decouple the influence of $\boldsymbol{\eta}(\tilde{\boldsymbol{x}}|\tilde{\boldsymbol{y}})$, under each $\tilde{y}$, in the inequality constraint. Finally, we replace the conditional EVaR definition by solving for $\boldsymbol{\eta}(\tilde{\boldsymbol{x}}|\tilde{\boldsymbol{y}})$ for a given $\boldsymbol{\zeta}$ in step (c), and representing $\boldsymbol{\eta}(\tilde{\boldsymbol{y}})$ using $\boldsymbol{\xi}$. □

The first part of our theorem follows directly from Proposition A.1. Suppose that $\alpha>0$; the result follows for $\alpha=0$ because $\mathrm{EVaR}_0\left[\cdot\right]$ reduces to $\mathrm{ess\,inf}$. Then, the second part of the corollary holds as

$$\mathrm{EVaR}_\alpha\left[r(\tilde{s},\tilde{a},\tilde{s}')\right] =$$

$$= \inf_{\boldsymbol{\zeta}\in(0,1]^N,\boldsymbol{\xi}\in\Delta_N}\left\{\sum_{s\in\mathcal{S}}\xi_s\,\mathrm{EVaR}_{\zeta_s}\left[r(s,\tilde{a},\tilde{s}')\mid\tilde{s}=s\right]\mid\sum_{s\in\mathcal{S}}\xi_s\log\frac{\xi_s}{\zeta_s\hat{p}_s}\leq-\log\alpha\right\}$$

$$\leq \inf_{\boldsymbol{\zeta}\in(0,1]^N,\boldsymbol{\xi}\in\Delta_N}\left\{\sum_{s\in\mathcal{S}}\xi_s\,\mathrm{EVaR}_{\zeta_s}\left[r(s,\tilde{a},\tilde{s}')\mid\tilde{s}=s\right]\mid\sum_{s\in\mathcal{S}}\xi_s\log\frac{\xi_s}{\zeta_s\hat{p}_s}\leq-\log\alpha,\,\boldsymbol{\xi}\leq\alpha^{-1}\hat{\boldsymbol{p}}\right\}$$

$$\leq \inf_{\boldsymbol{\xi}\in\Delta_N}\left\{\sum_{s\in\mathcal{S}}\xi_s\,\mathrm{EVaR}_{\alpha\xi_s\hat{p}_s^{-1}}\left[r(s,\tilde{a},\tilde{s}')\mid\tilde{s}=s\right]\mid\boldsymbol{\xi}\leq\alpha^{-1}\hat{\boldsymbol{p}}\right\}.$$

The first inequality follows from adding a constraint on the pairs on the $\boldsymbol{\xi}$ considered by the infimum. The second inequality follows by fixing $\zeta_s=\hat{\zeta}_s$ with $\hat{\zeta}_s=\alpha\xi_s\hat{p}_s^{-1}$ for each $s\in\mathcal{S}$. This is an upper bound because $\hat{\zeta}_s$ is feasible in the infimum:

$$\sum_{s\in\mathcal{S}}\xi_s\log\frac{\xi_s}{\hat{\zeta}_s\hat{p}_s} = -\log\alpha\leq-\log\alpha.$$

The value $\hat{\zeta}_s$ is well-defined since $\hat{p}_s>0$ and the constraint $\boldsymbol{\xi}\leq\alpha^{-1}\hat{\boldsymbol{p}}$ ensures that $\hat{\zeta}_s\leq1$. Also, we can relax the constraint $\zeta_s>0\Rightarrow\xi_s>0$ to $\xi_s\geq0$ because $\mathrm{EVaR}_0\left[\tilde{x}\right]=\lim_{\alpha\to0}\mathrm{EVaR}_\alpha\left[\tilde{x}\right]$, and, therefore, the infimum is not affected. Finally, the inequality in the corollary follows immediately by further upper bounding the decomposition above by adding a constraint. □

### A.4 Proof of Theorem 5.2

The equality develops from Theorem 5.1 as

$$
\begin{aligned}
\max_{\pi \in \Pi} \mathrm{VaR}_{\alpha}^{\tilde{a} \sim \boldsymbol{\pi}(\tilde{s})}[r(\tilde{s}, \tilde{a}, \tilde{s}')] &= \max_{\pi \in \Pi} \sup_{\boldsymbol{\zeta} \in \Delta_S : \alpha \cdot \boldsymbol{\zeta} \le \hat{\boldsymbol{p}}} \min_{s \in \mathcal{S}} \left( \mathrm{VaR}_{\alpha \zeta_s \hat{p}_s^{-1}}^{\tilde{a} \sim \boldsymbol{\pi}(s)}[r(s, \tilde{a}, \tilde{s}') \mid \tilde{s} = s] \right) \\
&= \sup_{\boldsymbol{\zeta} \in \Delta_S : \alpha \cdot \boldsymbol{\zeta} \le \hat{\boldsymbol{p}}} \max_{\pi \in \Pi} \min_{s \in \mathcal{S}} \left( \mathrm{VaR}_{\alpha \zeta_s \hat{p}_s^{-1}}^{\tilde{a} \sim \boldsymbol{\pi}(s)}[r(s, \tilde{a}, \tilde{s}') \mid \tilde{s} = s] \right) \\
&= \sup_{\boldsymbol{\zeta} \in \Delta_S : \alpha \cdot \boldsymbol{\zeta} \le \hat{\boldsymbol{p}}} \min_{s \in \mathcal{S}} \left( \max_{\boldsymbol{d} \in \Delta_A} \mathrm{VaR}_{\alpha \zeta_s \hat{p}_s^{-1}}^{\tilde{a} \sim \boldsymbol{d}}[r(s, \tilde{a}, \tilde{s}') \mid \tilde{s} = s] \right),
\end{aligned}
$$

where we first change the order of maximum and supremum, followed by changing the order of $\max_\pi \min_s$ with $\min_s \max_\pi$. The latter is a direct consequence of the interchangeability property of the maximum operation (Shapiro, 2017, proposition 2.2). $\qquad\square$

### A.5 Proof of Theorem 5.3

The proof mainly relies on correcting the decomposition of lower quantile proposed in (Li et al., 2022).

**Proposition A.2.** *Given an $\tilde{x} \in \mathbb{X}$, suppose that a random variable $\tilde{y} \colon \Omega \to \mathcal{N} = \{1, \dots, N\}$ is distributed as $\hat{\boldsymbol{p}} = (\hat{p}_i)_{i=1}^N$ with $\hat{p}_i > 0$. Then:*

$$
Q_\alpha(\tilde{x}) = \sup_{\boldsymbol{\zeta} \in [0,1]^N} \left\{ \min_{i \in \mathcal{N} : \zeta_i < 1} Q_{\zeta_i}(\tilde{x} \mid \tilde{y} = i) \mid \sum_{i=1}^N \zeta_i \hat{p}_i < \alpha \right\}, \tag{24}
$$

*where we interpret the supremum to be minus infinity if its feasible set is empty, which only occurs if $\alpha = 0$.*

*Proof.* First, we decompose the lower quantile using its definition as

$$
\begin{aligned}
Q_\alpha(\tilde{x}) = \sup\{z \mid \mathbb{P}[\tilde{x} < z] < \alpha\} &\stackrel{(a)}{=} \sup_{z \in \mathbb{R}} \left\{ z \mid \sum_{i=1}^N \mathbb{P}[\tilde{x} < z \mid \tilde{y} = i]\,\hat{p}_i < \alpha \right\} \\
&\stackrel{(b)}{=} \sup_{z \in \mathbb{R}, \boldsymbol{\zeta} \in [0,1]^N} \left\{ z \mid \sum_{i=1}^N \zeta_i \hat{p}_i < \alpha, \ \mathbb{P}[\tilde{x} < z \mid \tilde{y} = i] < \zeta_i, \forall i \in \mathcal{N} : \zeta_i < 1 \right\} \\
&\stackrel{(c)}{=} \sup_{z \in \mathbb{R}, \boldsymbol{\zeta} \in [0,1]^N} \left\{ z \mid z < Q_{\zeta_i}(\tilde{x} \mid \tilde{y} = i), \forall i \in \mathcal{N} : \zeta_i < 1, \ \sum_{i=1}^N \zeta_i \hat{p}_i < \alpha \right\} \\
&\stackrel{(d)}{=} \sup_{\boldsymbol{\zeta} \in [0,1]^N} \left\{ \sup_{z \in \mathbb{R}} \{ z \mid z < Q_{\zeta_i}(\tilde{x} \mid \tilde{y} = i), \forall i \in \mathcal{N} : \zeta_i < 1 \} \mid \sum_{i=1}^N \zeta_i \hat{p}_i < \alpha \right\} \\
&\stackrel{(e)}{=} \sup_{\boldsymbol{\zeta} \in [0,1]^N} \left\{ \min_{i \in \mathcal{N} : \zeta_i < 1} Q_{\zeta_i}(\tilde{x} \mid \tilde{y} = i) \mid \sum_{i=1}^N \zeta_i \hat{p}_i < \alpha \right\}.
\end{aligned}
$$

We decompose the probability $\mathbb{P}[\tilde{x} < z]$ in terms of the conditional probabilities $\mathbb{P}[\tilde{x} < z \mid \tilde{y} = i]$ in step (a) and then lower-bound them by an auxiliary variable $\zeta_i$ in step (b). In step (c), we exploit the following equivalence:

$$
\mathbb{P}[\tilde{x} < z \mid \tilde{y} = i] < \zeta_i \quad \Leftrightarrow \quad z < Q_{\zeta_i}(\tilde{x} \mid \tilde{y} = i)
$$

The direction $\Leftarrow$ in the equivalence follows from the definition of $Q_{\zeta_i}(\tilde{x} \mid \tilde{y} = i)$:

$$
z < Q_{\zeta_i}(\tilde{x} \mid \tilde{y} = i) = \inf\{z \mid \mathbb{P}[\tilde{x} \le z \mid \tilde{y} = i] \ge \zeta_i\} \Rightarrow \mathbb{P}[\tilde{x} < z \mid \tilde{y} = i] < \zeta_i.
$$

The direction $\Rightarrow$ follows from the definition of VaR (see equation (1)), which implies that VaR upper-bounds any $z$ that satisfies the left-hand condition:

$$
\mathbb{P}[\tilde{x} < z \mid \tilde{y} = i] < \zeta_i \Rightarrow Q_{\zeta_i}(\tilde{x} \mid \tilde{y} = i) = \sup\ \{z \in \mathbb{R} \mid \mathbb{P}[\tilde{x} < z \mid \tilde{y} = i] < \zeta_i\} \ge z,
$$

yet $Q_{\zeta_i}(\tilde{x} \mid \tilde{y} = i) \ne z$ otherwise since $\mathbb{P}[\tilde{x} < z \mid \tilde{y} = i]$ is right continuous, there must exist some $\epsilon > 0$ for which $\mathbb{P}[\tilde{x} < z + \epsilon \mid \tilde{y} = i] < \zeta_i$ hence:

$$
z = \sup\ \{z \in \mathbb{R} \mid \mathbb{P}[\tilde{x} < z \mid \tilde{y} = i] < \zeta_i\} \ge z + \epsilon > z,
$$

which leads to a contradiction. In step (e), we solve for $z$. Finally, we obtain the form in (24). $\qquad\square$

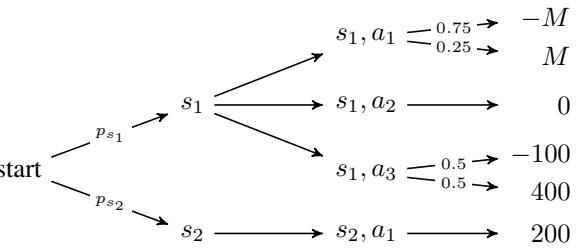

Figure 4: An example used to show the sub-optimality of a policy in Appendix B.

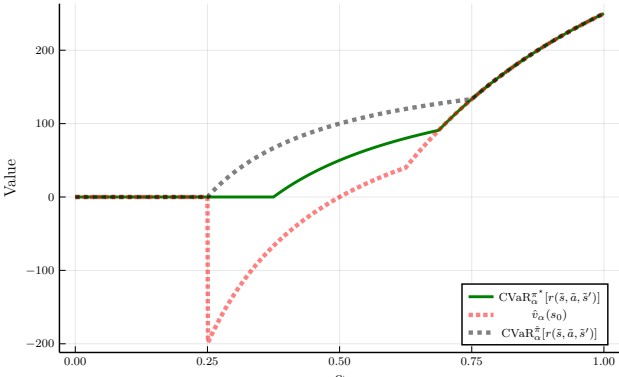

Figure 5: Return computed in Chow et al. (2015) vs optimal CVaR policy for the example in Figure 4 with $p_{s_1} = p_{s_2} = 0.5$ and $M = 600$. The optimal CVaR policy for each $\alpha$ is denoted by $\pi^\star$. The value function $\hat{v}$ is computed according to the decomposition in the r.h.s. of (8) and the corresponding policy is $\hat{\pi}$.

## B  CVaR Suboptimal Policy

In this section, we construct a simple MDP to demonstrate that the suboptimality of the CVaR decomposition discussed in Section 3 can also lead to computing a suboptimal policy. First, we give a particular example in which the policy computed according to Chow et al. (2015) is suboptimal. Then, we show that the sub-optimality gap can be arbitrarily large.

To demonstrate the suboptimality of the policy computed in Chow et al. (2015), consider an MDP with two states and three actions and a horizon $T = 1$ as shown in Figure 4. Also let $M = 600$ and

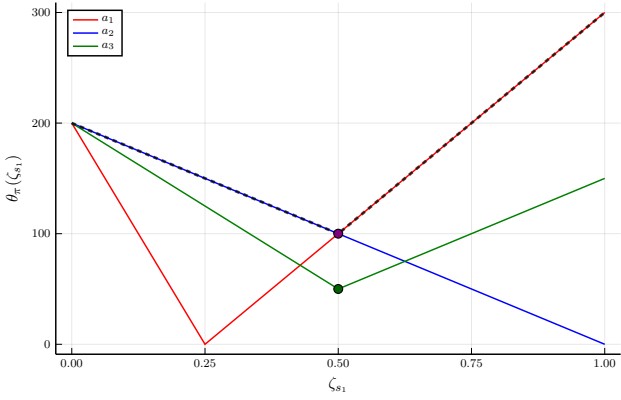

Figure 6: The function $\theta_\pi(\cdot)$ and $\zeta_{s1}$ in CVaR decomposition at $\alpha = 0.5$ for the example in Figure 4 with $p_{s_1} = p_{s_2} = 0.5$ and $M = 600$.

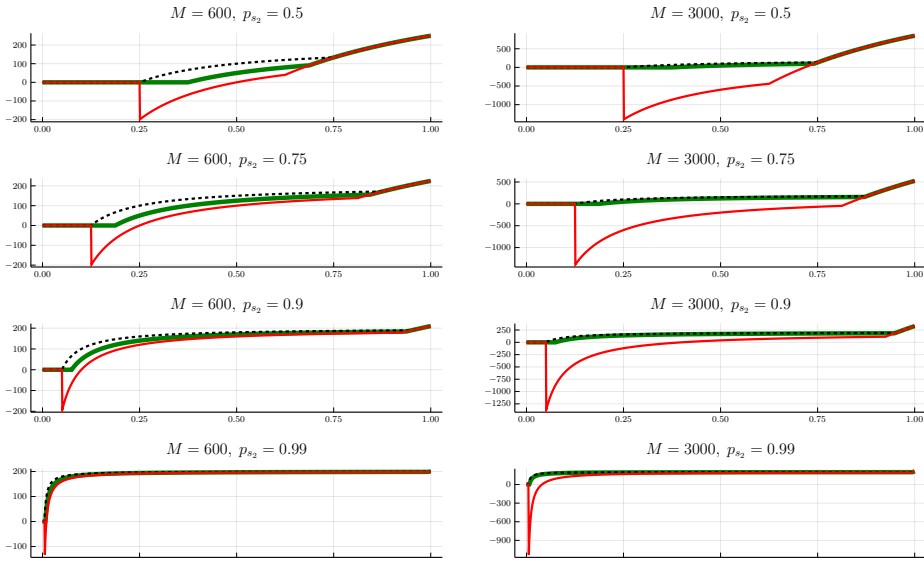

Figure 7: CVaR decomposition suboptimality for different $M$ and $p_{s2}$ for MDP defined in Figure 4

$p_{s_1} = p_{s_2} = 0.5$. An optimal solution for the $s_1$ sub-problem in this example is

$$\text{CVaR}_{\alpha'}[r(\tilde{s}) \mid \tilde{s} = s_1] = \begin{cases} \text{CVaR}_{\alpha'}[r(s_1, a_2, \tilde{s}')] & \text{if } \alpha' < 0.5 \\ \text{CVaR}_{\alpha'}[r(s_1, a_1, \tilde{s}')] & \text{if } \alpha' \geq 0.5. \end{cases}$$

Figure 5 depicts the sub-optimality demonstrated by the CVaR decomposition. CVaR decomposition for policy optimization would over-approximate (black dash) the value function and commit to a sub-optimal solution (red solid). This simple example answers the following important concerns on the suboptimality.

First, CVaR policy optimization decomposition in Chow et al. (2015) can lead one to choose a suboptimal policy . In particular, Figure 5 shows that the CVaR policy optimization decomposition is an overestimate of the return and the decision maker commits to the worst action for CVaR objective $\alpha \in (0.25, 625)$. Furthermore, action $a_3$ is never chosen for (Chow et al., 2015) even though it is the only optimal action for $\alpha \in (0.375, 0.6875)$. We can see from Figure 6 when we inspect the $\zeta$ function for $\alpha = 0.5$ that $a_3$ is optimal with CVaR return of 50. However, the CVaR decomposition would select $a_1$ or $a_2$ and over-estimate the CVaR return to 100, but the sub-optimal actions $a_1$ or $a_2$ would have the true CVaR return of 0 instead.

Second, there is no upper bound on the sub-optimality of the action chosen with respect CVaR return of the optimal action. The regret of the sub-optimal return is dependent on the distribution of the sub-optimal action and unbounded with respect to the true optimal return, as the performance of the sub-optimal return for $\alpha \in (0.25, 0.5)$ is worse than the worst case scenario of the best worst-case policy. Increasing the magnitude ($M > 400$) for the reward instead of the sub-optimal action $a_1$ in $s_1$ in Figure 4 increases the sub-optimality and can be arbitrarily large.

Third, the sub-optimality can occur for any $\alpha \in (0, 1)$. As above, we use the MDP in Figure 4 and perturb two values to demonstrate the suboptimality by (i) increasing the magnitude ($M > 400$) for the reward of the sub-optimal action $s_1, a_1$, or (ii) increasing the initial state probability ($p_{s_2}$). That leads to increasing the range where the CVaR decomposition for policy optimization yields a sub-optimal policy as shown in Figure 7.

# C Extension to Longer Horizon

For completeness, we include in this section an extension of the corrected decomposition presented in Theorem 5.2 to horizon $T > 1$. Our demonstration establishes that the decomposition proposed in Theorem 1 and 3 of (Li et al., 2022) actually applies to the upper quantile (i.e. value-at-risk) of the cumulated reward rather than the lower one as claimed by the authors. In comparison with the proof found in (Li et al., 2022), we pay close attention to demonstrating that the supremum in (5.1) is attained when finite and that deterministic policies are optimal.

Given any finite MDP with horizon $T > 0$ and $\alpha \in [0, 1]$, we define the initial value as

$$v_0 := \max_{\boldsymbol{\zeta} \in \Delta_S} \left\{ \min_{s \in \mathcal{S}} \max_{a \in \mathcal{A}} q_1(s, a, \alpha \zeta_s \hat{p}_s^{-1}) \mid \alpha \cdot \boldsymbol{\zeta} \leq \hat{\boldsymbol{p}} \right\}. \tag{25}$$

The state-action value function $q_t : \mathcal{S} \times \mathcal{A} \times [0, 1] \to \mathbb{R}$ for the terminal step $t = T + 1$ and $\alpha \in [0, 1]$ and each $s \in \mathcal{S}$ and $a \in \mathcal{A}$ is defined as

$$q_{T+1}(s, a, \alpha) := \begin{cases} \infty & \text{if } \alpha = 1 \\ 0 & \text{otherwise.} \end{cases}$$

For each $t = 1, \ldots, T$ and $\alpha < 1$ the state-action function is defined recursively as

$$q_t(s, a, \alpha) := \max_{\boldsymbol{\zeta} \in \Delta_S} \left\{ \min_{s' \in \mathcal{S}} \max_{a' \in \mathcal{A}} r(s, a, s') + q_{t+1}(s', a', \alpha \zeta_{s'} p_{sas'}^{-1}) \mid \alpha \cdot \boldsymbol{\zeta} \leq \boldsymbol{p}_{sa} \right\}. \tag{26}$$

For $\alpha = 1$, the state-action value function is defined as

$$q_t(s, a, 1) := \infty.$$

To construct the optimal policy, we also need to define a history-dependent risk level $\bar{\alpha}_t : \mathcal{S}^t \times \mathcal{A}^{t-1} \to [0, 1]$ that satisfies for each $t = 1, \ldots, T$ and $s_1, \ldots, s_t$ and $a_1, \ldots, a_{t-1}$ that

$$q_t(s, a, \bar{\alpha}_t(s_{1:t}, a_{1:t-1})) = \min_{s' \in \mathcal{S}} \max_{a' \in \mathcal{A}} r(s, a, s') + q_{t+1}(s', a', \bar{\alpha}_{t+1}([s_{1:t}, s'], [a_{1:t-1}, a'])).$$

The appropriate values $\bar{\alpha}_t$ can be readily recovered from the optimal solution to (26). Finally, letting

$$a_t^*(s_t, \alpha_t) \in \begin{cases} \mathcal{A} & \text{if } \alpha_t = 1 \\ \arg\max_{a \in \mathcal{A}} q_t(s_t, a, \alpha_t) & \text{otherwise },\end{cases} \tag{27}$$

we construct a *deterministic* history-dependent policy $\bar{\pi}_t : \mathcal{S}^t \times \mathcal{A}^{t-1} \to \mathcal{A}$ for each $t = 1, \ldots, T$ and $s_1, \ldots, s_t$ and $a_1, \ldots, a_{t-1}$ that puts all the probability mass on $a_t^*(s_t, \bar{\alpha}_t(s_{1:t}, a_{1:t-1}))$, i.e.

$$\mathbb{P}^{\tilde{a}_t \sim \bar{\pi}_t(s_{1:t}, a_{1:t-1})}(\tilde{a}_t = a_t^*(s_t, \bar{\alpha}_t(s_{1:t}, a_{1:t-1}))) = 1. \tag{28}$$

The following theorem states that the value function defined above represents the VaR of the returns of the optimal policy. Moreover, the history-dependent policy $\bar{\pi}$ above is optimal and attains the optimal VaR return.

**Theorem C.1.** *For a finite horizon $T$ and any $\alpha \in [0, 1]$ we have that $v_0$ defined in (25) coincides with the optimal VaR return:*

$$v_0 = \max_{\pi \in \Pi} \mathrm{VaR}_\alpha^{\tilde{a}_t \sim \boldsymbol{\pi}_t(\tilde{s}_{1:t}, \tilde{a}_{1:t-1})} \left[ \sum_{t=1}^T r(\tilde{s}_t, \tilde{a}_t, \tilde{s}_{t+1}) \right].$$

*Moreover, this optimal return is attained by the policy $(\bar{\pi}_t)_{t=1}^T$ defined in (27):*

$$v_0 = \mathrm{VaR}_\alpha^{\tilde{a}_t \sim \bar{\boldsymbol{\pi}}_t(\tilde{s}_{1:t}, \tilde{a}_{1:t-1})} \left[ \sum_{t=1}^T r(\tilde{s}_t, \tilde{a}_t, \tilde{s}_{t+1}) \right].$$

In proving Theorem C.1, we will make use of the upper semicontinuity property of $\mathrm{VaR}_\alpha[\tilde{x}]$, which is presented in the following Lemma for completeness.

**Lemma C.2.** *The function $\mathrm{VaR}_\alpha[\tilde{x}]$ is upper semicontinuous in $\alpha$ on the interval $\alpha \in [0, 1]$.*

*Proof.* This follows from the fact that $\text{VaR}_\alpha[\tilde{x}]$ is non-decreasing and right-continuous in terms of $\alpha$ (see Lemma A.19 in (Follmer and Schied, 2016)). □

*Proof of Theorem C.1.* The proof consists of two steps addressing the two equalities stated by the theorem.

**Step 1:** For all $t = 1, \ldots, T$, we let

$$\hat{q}_t(s_t, a_t, \alpha_t) :=$$

$$\max_{\pi \in \Pi_{t+1:T}} \text{VaR}_{\alpha_t}^{\tilde{a}_{t'} \sim \boldsymbol{\pi}_{t'}(\tilde{s}_{t+1:t'}, \tilde{a}_{t+1:t'-1})} \left[ r(s_t, a_t, \tilde{s}_{t+1}) + \sum_{t'=t+1}^{T} r(\tilde{s}_{t'}, \tilde{a}_{t'}, \tilde{s}_{t'+1}) \mid s_t, a_t \right]$$

where $\Pi_{t+1:T}$ refers to the set of all history-dependent policies starting from time $t + 1$, and where $\text{VaR}_\alpha[\tilde{x}|s_t, a_t]$ is short for $\text{VaR}_\alpha[\tilde{x}|\tilde{s}_t = s_t, \tilde{a}_t = a_t]$. We will first show how $\hat{q}_t(s, a, \alpha) = q_t(s, a, \alpha)$ for all $t = 1, \ldots, T$, $s \in \mathcal{S}$, $a \in \mathcal{A}$, and $\alpha \in [0, 1]$. We follow with our conclusion about $v_0$.

First, addressing the case $\alpha = 1$, one can easily see that for all $t = 1, \ldots, T$, $s \in \mathcal{S}$, and $a \in \mathcal{A}$, we have that

$$\hat{q}_t(s_t, a_t, 1) = \max_{\pi \in \Pi_{t+1:T}} \text{VaR}_1^{\tilde{a}_{t'} \sim \boldsymbol{\pi}_{t'}(\tilde{s}_{t+1:t'}, \tilde{a}_{t+1:t'-1})} \left[ r(s_t, a_t, \tilde{s}_{t+1}) + \sum_{t'=t+1}^{T} r(\tilde{s}_{t'}, \tilde{a}_{t'}, \tilde{s}_{t'+1}) \mid s_t, a_t \right]$$

$$= \infty = q_t(s_t, a_t, 1),$$

based on our definition of value-at-risk for $\alpha = 1$.

Next, for $0 \leq \alpha < 1$ and using $\boldsymbol{p}$ short for $\boldsymbol{p}_{sa}$ when can start looking at time $t = T$, where we have that for all $s_T \in \mathcal{S}$, and $a_T \in \mathcal{A}$ :

$$\hat{q}_T(s_T, a_T, \alpha_T) = \text{VaR}_{\alpha_T}[r(s_T, a_T, \tilde{s}_{t+1})|s_T, a_T]$$

$$= \sup_{\boldsymbol{\zeta} \in \Delta_S} \left\{ \min_{s' \in \mathcal{S}} \text{VaR}_{\alpha_T \zeta_{s'} p_{s'}^{-1}}[r(s_T, a_T, s')] \mid \alpha_T \cdot \boldsymbol{\zeta} \leq \boldsymbol{p} \right\}$$

$$= \max_{\boldsymbol{\zeta} \in \Delta_S} \left\{ \min_{s' \in \mathcal{S}} \text{VaR}_{\alpha_T \zeta_{s'} p_{s'}^{-1}}[r(s_T, a_T, s')] \mid \alpha_T \cdot \boldsymbol{\zeta} \leq \boldsymbol{p} \right\}$$

$$= \max_{\boldsymbol{\zeta} \in \Delta_S} \left\{ \min_{s' \in \mathcal{S}} r(s_T, a_T, s') + \text{VaR}_{\alpha_T \zeta_s p_s^{-1}}[0] \mid \alpha_T \cdot \boldsymbol{\zeta} \leq \boldsymbol{p} \right\}$$

$$= \max_{\boldsymbol{\zeta} \in \Delta_S} \left\{ \min_{s' \in \mathcal{S}} r(s_T, a_T, s') + \max_{a' \in \mathcal{A}} q_{T+1}(s', a', \alpha_T \zeta_{s'} p_{s'}^{-1}) \mid \alpha_T \cdot \boldsymbol{\zeta} \leq \boldsymbol{p} \right\}$$

$$= q_T(s_T, a_T, \alpha_T),$$

where we first use the definition of $\hat{q}_T(s, a, \alpha)$, then the decomposition of VaR in Theorem 5.1. We follow with confirming, based on extreme value theory, that the supremum over $\boldsymbol{\zeta}$ must be achieved given that $\text{VaR}_\alpha(\tilde{x})$ of a random variable $\tilde{x}$ is upper semi-continuous (usc) in $\alpha$ (see Lemma C.2), that the minimum of a finite set of usc functions is usc, and that $\Delta_S$ is compact. The other two steps follow from the translation invariance of VaR and the definition of $q_{T+1}(s, a, \alpha)$.

Moreover, for all $1 \leq t < T$ let $\tilde{h}_{t:t'} := (\tilde{s}_{t:t'}, \tilde{a}_{t:t'-1})$ as the history between $t$ and $t'$. Using the short-hand $\tilde{r}_{t'} = r(\tilde{s}_{t'}, \tilde{a}_{t'}, \tilde{s}_{t'+1})$, and when $0 \leq \alpha_t < 1$, we have that:

$$\hat{q}_t(s_t, a_t, \alpha_t) = \max_{\pi \in \Pi_{t+1:T}} \text{VaR}_{\alpha_t}^{\tilde{a}_{t'} \sim \boldsymbol{\pi}_{t'}(\tilde{h}_{t+1:t'})} \left[ r(s_t, a_t, \tilde{s}_{t+1}) + \sum_{t'=t+1}^{T} r(\tilde{s}_{t'}, \tilde{a}_{t'}, \tilde{s}_{t'+1})|s_t, a_t \right]$$

$$= \max_{\pi \in \Pi_{t+1:T}} \sup_{\boldsymbol{\zeta} \in \Delta_S} \left\{ \min_{s' \in \mathcal{S}} \text{VaR}_{\alpha_t \zeta_{s'} p_{s'}^{-1}}^{\tilde{a}_{t'} \sim \boldsymbol{\pi}_{t'}(\tilde{h}_{t+1:t'})} \left[ r(s_t, a_t, s') + \sum_{t'=t+1}^{T} \tilde{r}_{t'} \mid \tilde{s}_{t+1} = s' \right] \mid \alpha_t \cdot \boldsymbol{\zeta} \leq \boldsymbol{p} \right\}$$

$$= \max_{\pi \in \Pi_{t+1:T}} \max_{\boldsymbol{\zeta} \in \Delta_S} \left\{ \min_{s' \in \mathcal{S}} \text{VaR}_{\alpha_t \zeta_{s'} p_{s'}^{-1}}^{\tilde{a}_{t'} \sim \boldsymbol{\pi}_{t'}(\tilde{h}_{t+1:t'})} \left[ r(s_t, a_t, s') + \sum_{t'=t+1}^{T} \tilde{r}_{t'} \mid \tilde{s}_{t+1} = s' \right] \mid \alpha_t \cdot \boldsymbol{\zeta} \leq \boldsymbol{p} \right\}$$

$$= \max_{\boldsymbol{\zeta} \in \Delta_S} \left\{ \max_{\pi \in \Pi_{t+1:T}} \min_{s' \in \mathcal{S}} \text{VaR}_{\alpha_t \zeta_{s'} p_{s'}^{-1}}^{\tilde{a}_{t'} \sim \boldsymbol{\pi}_{t'}(\tilde{h}_{t+1:t'})} [r(s_t, a_t, s') + \sum_{t'=t+1}^{T} \tilde{r}_{t'} \mid \tilde{s}_{t+1} = s'] \mid \alpha_t \cdot \boldsymbol{\zeta} \leq \boldsymbol{p} \right\}$$

$$= \max_{\boldsymbol{\zeta} \in \Delta_S} \left\{ \min_{s' \in \mathcal{S}} \max_{\boldsymbol{d} \in \Delta_A, \{\pi^a\}_{a \in \mathcal{A}} \in \Pi_{t+2:T}^{|\mathcal{A}|}} \text{VaR}_{\alpha_t \zeta_{s'} p_{s'}^{-1}}^{\tilde{a}_{t+1} \sim \boldsymbol{d}, \tilde{a}_{t'} \sim \boldsymbol{\pi}_{t'}^{\tilde{a}_{t+1}}(\tilde{h}_{t+2:t'})} [r(s_t, a_t, s') + \right.$$

$$\left. \sum_{t'=t+1}^{T} \tilde{r}_{t'} \mid \tilde{s}_{t+1} = s'] \mid \alpha_t \cdot \boldsymbol{\zeta} \leq \boldsymbol{p} \right\}$$

$$= \max_{\boldsymbol{\zeta} \in \Delta_S} \left\{ \min_{s' \in \mathcal{S}} r(s_t, a_t, s') + \right.$$

$$\left. \max_{\boldsymbol{d} \in \Delta_A, \{\pi^a\}_{a \in \mathcal{A}} \in \Pi_{t+2:T}^{|\mathcal{A}|}} \text{VaR}_{\alpha_t \zeta_{s'} p_{s'}^{-1}}^{\tilde{a}_{t+1} \sim \boldsymbol{d}, \tilde{a}_{t'} \sim \boldsymbol{\pi}_{t'}^{\tilde{a}_{t+1}}(\tilde{h}_{t+2:t'})} [\sum_{t'=t+1}^{T} \tilde{r}_{t'} \mid \tilde{s}_{t+1} = s'] \mid \alpha_t \cdot \boldsymbol{\zeta} \leq \boldsymbol{p} \right\}$$

$$= \max_{\boldsymbol{\zeta} \in \Delta_S} \left\{ \min_{s' \in \mathcal{S}} r(s_t, a_t, s') + \hat{v}_{t+1}(s', \alpha_t \zeta_{s'} p_{s'}^{-1}) \mid \alpha_t \cdot \boldsymbol{\zeta} \leq \boldsymbol{p} \right\} \tag{29}$$

where, as before, we start by exploiting the decomposition Theorem 5.1 and explicating that the supremum is attained. We then change the order of the two maximums, followed by changing the order of $\max_\pi \min_{s'}$ with $\min_{s'} \max_\pi$, which follows based on the interchangeability property of the minimum operation (Shapiro, 2017, proposition 2.2). We finally introduced a $(s, \alpha)$-value function operator which can be reduced as follows:

$$\hat{v}_t(s, \alpha) := \max_{\boldsymbol{d} \in \Delta_A, \{\pi^a\}_{a \in \mathcal{A}} \in \Pi_{t+1:T}^{|\mathcal{A}|}} \text{VaR}_\alpha^{\tilde{a}_t \sim \boldsymbol{d}, \tilde{a}_{t'} \sim \boldsymbol{\pi}_{t'}^{\tilde{a}_t}(\tilde{h}_{t+1:t})} [\sum_{t'=t}^{T} r(\tilde{s}_{t'}, \tilde{a}_{t'}, \tilde{s}_{t'+1}) \mid \tilde{s}_t = s]$$

$$= \max_{\{\pi^a\}_{a \in \mathcal{A}} \in \Pi_{t+1:T}^{|\mathcal{A}|}} \max_{\boldsymbol{d} \in \Delta_A} \text{VaR}_\alpha^{\tilde{a}_t \sim \boldsymbol{d}, \tilde{a}_{t'} \sim \boldsymbol{\pi}_{t'}^{\tilde{a}_t}(\tilde{h}_{t+1:t})} [\sum_{t'=t}^{T} r(\tilde{s}_{t'}, \tilde{a}_{t'}, \tilde{s}_{t'+1}) \mid \tilde{s}_t = s']$$

$$= \max_{\{\pi^a\}_{a \in \mathcal{A}} \in \Pi_{t+1:T}^{|\mathcal{A}|}} \max_{a \in \mathcal{A}} \text{VaR}_\alpha^{\tilde{a}_{t'} \sim \boldsymbol{\pi}_{t'}^{a}(\tilde{h}_{t+1:t})} [r(s, a, \tilde{s}_{t+1}) + \sum_{t'=t+1}^{T} r(\tilde{s}_{t'}, \tilde{a}_{t'}, \tilde{s}_{t'+1}) \mid \tilde{s}_t = s, \tilde{a}_t = a]$$

$$= \max_{a \in \mathcal{A}} \max_{\pi \in \Pi_{t+1:T}} \text{VaR}_\alpha^{\tilde{a}_{t'} \sim \boldsymbol{\pi}_{t'}(\tilde{h}_{t+1:t'})} [r(s, a, \tilde{s}_{t+1}) + \sum_{t'=t+1}^{T} r(\tilde{s}_{t'}, \tilde{a}_{t'}, \tilde{s}_{t'+1}) \mid \tilde{s}_t = s, \tilde{a}_t = a]$$

$$= \max_{a \in \mathcal{A}} \hat{q}_t(s, a, \alpha), \tag{30}$$

where we exploited the fact that value-at-risk is a mixture quasi-convex function (Delage et al., 2019), meaning that it cannot be improved by a randomized policy.

Hence, replacing $\hat{v}$ back in equation (29), we get

$$\hat{q}_t(s_t, a_t, \alpha_t) = \max_{\boldsymbol{\zeta} \in \Delta_S} \left\{ \min_{s' \in \mathcal{S}} r(s_t, a_t, s') + \max_{a \in \mathcal{A}} \hat{q}_{t+1}(s', \alpha_t \zeta_{s'} p_{s'}^{-1}) \mid \alpha_t \cdot \boldsymbol{\zeta} \leq \boldsymbol{p} \right\}$$

$$= \max_{\boldsymbol{\zeta} \in \Delta_S} \left\{ \min_{s' \in \mathcal{S}} r(s_t, a_t, s') + \max_{a \in \mathcal{A}} q_{t+1}(s', \alpha_t \zeta_{s'} p_{s'}^{-1}) \mid \alpha_t \cdot \boldsymbol{\zeta} \leq \boldsymbol{p} \right\}$$

$$= q_t(s_t, a_t, \alpha_t).$$

Finalizing our conclusion, we get using the short-hand $\tilde{r}_t = r(\tilde{s}_t, \tilde{a}_t, \tilde{s}_{t+1})$ that

$$\max_{\pi \in \Pi} \text{VaR}_\alpha^{\tilde{a}_t \sim \boldsymbol{\pi}_t(\tilde{h}_{1:t})} \left[ \sum_{t=1}^{T} \tilde{r}_t \right] =$$

$$= \max_{\pi \in \Pi} \sup_{\boldsymbol{\zeta} \in \Delta_S} \left\{ \min_{s' \in \mathcal{S}} \text{VaR}_{\alpha \zeta_{s'} \hat{p}_{s'}^{-1}}^{\tilde{a}_t \sim \boldsymbol{\pi}_t(\tilde{h}_{1:t})} \left[ \sum_{t=1}^{T} \tilde{r}_t \mid \tilde{s}_1 = s' \right] \mid \alpha \cdot \boldsymbol{\zeta} \leq \hat{\boldsymbol{p}} \right\}$$

$$= \sup_{\boldsymbol{\zeta} \in \Delta_S} \left\{ \min_{s' \in \mathcal{S}} \max_{\boldsymbol{d} \in \Delta_A, \{\pi^a\}_{a \in \mathcal{A}} \in \Pi_{2:T}^{|\mathcal{A}|}} \text{VaR}_{\alpha \zeta_{s'} \hat{p}_{s'}^{-1}}^{\tilde{a}_1 \sim \boldsymbol{d}, \tilde{a}_t \sim \boldsymbol{\pi}_t^{\tilde{a}_1}(\tilde{h}_{2:t})} \left[ \sum_{t=1}^{T} \tilde{r}_t \mid \tilde{s}_1 = s' \right] \mid \alpha \cdot \boldsymbol{\zeta} \leq \hat{\boldsymbol{p}} \right\}$$

$$= \sup_{\boldsymbol{\zeta} \in \Delta_S} \left\{ \min_{s' \in \mathcal{S}} \max_{a \in \mathcal{A}, \pi \in \Pi_{2:T}} \mathrm{VaR}_{\alpha \zeta_{s'} \hat{p}_{s'}^{-1}}^{\tilde{a}_t \sim \boldsymbol{\pi}(\tilde{h}_{2:t})} \left[ \sum_{t=1}^{T} \tilde{r}_t \mid \tilde{s}_1 = s' \right] \mid \alpha \cdot \boldsymbol{\zeta} \le \hat{\boldsymbol{p}} \right\}$$

$$= \sup_{\boldsymbol{\zeta} \in \Delta_S} \left\{ \min_{s' \in \mathcal{S}} \max_{a \in \mathcal{A}} \hat{q}_1(s', a, \alpha \zeta_{s'} \hat{p}_{s'}^{-1}) \mid \tilde{s}_1 = s' \right] \mid \alpha \cdot \boldsymbol{\zeta} \le \hat{\boldsymbol{p}} \right\}$$

$$= \sup_{\boldsymbol{\zeta} \in \Delta_S} \left[ \min_{s' \in \mathcal{S}} \max_{a \in \mathcal{A}} q_1(s', a, \alpha \zeta_{s'} \hat{p}_{s'}^{-1}) \mid \tilde{s}_1 = s' \right] \mid \alpha \cdot \boldsymbol{\zeta} \le \hat{\boldsymbol{p}} \right]$$

$$= v_0 \,,$$

where we employ the same steps as for reducing $\hat{q}_t$ to $q_t$.

**Step 2:** Regarding the optimality of the proposed policy, we will show inductively that

$$\bar{\bar{q}}_t(s_t, a_t, \alpha_t) :=$$

$$\mathrm{VaR}_{\alpha_t}^{\tilde{a}_{t'} \sim \bar{\boldsymbol{\pi}}_{t'}(\tilde{h}_{1:t'})} \left[ r(s_t, a_t, \tilde{s}_{t+1}) + \sum_{t'=t+1}^{T} r(\tilde{s}_{t'}, \tilde{a}_{t'}, \tilde{s}_{t'+1}) \mid s_t, a_t, \bar{\alpha}_t(\tilde{s}_{1:t}, \tilde{a}_{1:t-1}) = \alpha_t \right]$$

is an upper envelope for $\hat{q}_t(s_t, a_t, \alpha_t)$ for all $1 \le t \le T$. Conditioning on $\alpha_t$ in the definition above is necessary because the policy $\hat{\boldsymbol{\pi}}$ depends on the risk level $\alpha$. This can then be exploited to obtain the following inequality:

$$\mathrm{VaR}_{\alpha}^{\tilde{a}_t \sim \bar{\boldsymbol{\pi}}_t(\tilde{h}_{1:t})} [\sum_{t=1}^{T} r(\tilde{s}_t, \tilde{a}_t, \tilde{s}_{t+1})] =$$

$$= \sup_{\boldsymbol{\zeta} \in \Delta_S} \left\{ \min_{s \in \mathcal{S}} \mathrm{VaR}_{\alpha \zeta_s \hat{p}_s}^{\tilde{a}_t \sim \bar{\boldsymbol{\pi}}_t(\tilde{h}_{1:t})} [\sum_{t=1}^{T} r(\tilde{s}_t, \tilde{a}_t, \tilde{s}_{t+1}) | \tilde{s}_1 = s] \mid \alpha \cdot \boldsymbol{\zeta} \le \hat{\boldsymbol{p}} \right\}$$

$$= \sup_{\boldsymbol{\zeta} \in \Delta_S} \left\{ \min_{s \in \mathcal{S}} \mathrm{VaR}_{\alpha \zeta_s \hat{p}_s}^{\tilde{a}_t \sim \bar{\boldsymbol{\pi}}_t(\tilde{h}_{1:t})} [\sum_{t=1}^{T} r(\tilde{s}_t, \tilde{a}_t, \tilde{s}_{t+1}) | \tilde{s}_1 = s, \tilde{a}_1 = a_1^*(s, \bar{\alpha}_1(s))] \mid \alpha \cdot \boldsymbol{\zeta} \le \hat{\boldsymbol{p}} \right\}$$

$$\ge \min_{s \in \mathcal{S}} \mathrm{VaR}_{\bar{\alpha}_1(s)}^{\tilde{a}_t \sim \bar{\boldsymbol{\pi}}_t(\tilde{h}_{1:t})} [\sum_{t=1}^{T} r(\tilde{s}_t, \tilde{a}_t, \tilde{s}_{t+1}) | \tilde{s}_1 = s, \tilde{a}_1 = a_1^*(s, \bar{\alpha}_1(s)), \bar{\alpha}_1(\tilde{s}_1) = \bar{\alpha}_1(s)]$$

$$= \min_{s \in \mathcal{S}} \bar{\bar{q}}_1(s, a_1^*(s, \bar{\alpha}_1(s)), \bar{\alpha}_1(s)) \; \ge \; \min_{s \in \mathcal{S}} \hat{q}_1(s, a_1^*(s, \bar{\alpha}_1(s)), \bar{\alpha}_1(s))$$

$$= \min_{s \in \mathcal{S}} q_1(s, a_1^*(s, \bar{\alpha}_1(s)), \bar{\alpha}_1(s)) \; = \; \min_{s \in \mathcal{S}} \max_{a \in \mathcal{A}} q_1(s, a, \bar{\alpha}_1(s))$$

$$= \max_{\boldsymbol{\zeta} \in \Delta_S} \left\{ \min_{s \in \mathcal{S}} \max_{a \in \mathcal{A}} q_1(s, a, \bar{\alpha}_1(s)) \mid \alpha \cdot \boldsymbol{\zeta} \le \hat{\boldsymbol{p}} \right\} = v_0$$

$$= \max_{\pi \in \Pi} \mathrm{VaR}_{\alpha}^{\tilde{a}_t \sim \boldsymbol{\pi}_t(\tilde{h}_{1:t})} \left[ \sum_{t=1}^{T} r(\tilde{s}_t, \tilde{a}_t, \tilde{s}_{t+1}) \right]$$

$$\ge \mathrm{VaR}_{\alpha}^{\tilde{a}_t \sim \bar{\boldsymbol{\pi}}_t(\tilde{h}_{1:t})} \left[ \sum_{t=1}^{T} r(\tilde{s}_t, \tilde{a}_t, \tilde{s}_{t+1}) \right] \,,$$

where the first step comes from the decomposition of VaR, the second from the definition of $\bar{\pi}_1$. The third step follows from the feasibility of $\zeta_s := \bar{\alpha}_1(s)/(\alpha \hat{p}_s^{-1})$. We then exploit the definition of $\bar{\bar{q}}_1$, the fact that it upper bounds $\hat{q}_1$, the equivalence of $\hat{q}_1$ and $q_1$, and definition of $a_1^*$. On the sixth line, we exploit the definition of $\bar{\alpha}_1(s)$, followed with the definition of $v_0$. These steps would therefore confirm that $\bar{\pi}$ is optimal.

We are left with showing that $\bar{\bar{q}}_t$ upper bounds $\hat{q}_t$ for all $t = 1, \ldots, T$. Specifically, starting with $t = T$, we have:

$$\bar{\bar{q}}_T(s_T, a_T, \alpha_T) = \mathrm{VaR}_{\alpha_T}[r(s_T, a_T, \tilde{s}_{T+1}) | s_T, a_T] = \hat{q}_T(s_T, a_T, \alpha_T) \tag{31}$$

Then, for $t < T$ we focus on the case $\alpha_t < 1$, since otherwise we have $\infty \geq \infty$. Specifically, we first define

$$\bar{v}_t(s_t, \alpha_t) := \text{VaR}_{\alpha_t}^{\tilde{a}_{t'} \sim \bar{\pi}_{t'}(\tilde{h}_{1:t'})} [r(s_t, \tilde{a}_t, \tilde{s}_{t+1}) + \sum_{t'=t+1}^{T} r(\tilde{s}_{t'}, \tilde{a}_{t'}, \tilde{s}_{t'+1}) | s_t, \bar{\alpha}_t(\tilde{s}_{1:t}, \tilde{a}_{1:t-1}) = \alpha_t],$$

which captures the value-at-risk at level $\alpha_t$ of the reward-to-go looking forward from time $t$ when running $\bar{\pi}$ given that the history up to time $t$ satisfies $\bar{\alpha}_t(\tilde{s}_{1:t}, \tilde{a}_{1:t-1}) = \alpha_t$.

We then take inductive steps starting from $t = T$ down to $t = 2$. At each $t$, we can verify that if $\bar{\hat{q}}_t$ is an upper envelope for $\hat{q}_t$, then :

$$\bar{v}_t(s_t, 1) = \infty \geq \infty = \max_{a \in \mathcal{A}} \hat{q}_t(s_t, a, 1) = \hat{v}_t(s_t, 1)$$

and that for $0 \leq \alpha_t < 1$:

$$\bar{v}_t(s_t, \alpha_t) = \text{VaR}_{\alpha_t}^{\tilde{a}_{t'} \sim \bar{\pi}_{t'}(\tilde{h}_{1:t'})} \Big[ r(s_t, \tilde{a}_t, \tilde{s}_{t+1})$$

$$+ \sum_{t'=t+1}^{T} r(\tilde{s}_{t'}, \tilde{a}_{t'}, \tilde{s}_{t'+1}) \mid s_t, \tilde{a}_t = a_t^*(s_t, \alpha_t), \bar{\alpha}_t(\tilde{s}_{1:t}, \tilde{a}_{1:t-1}) = \alpha_t \Big]$$

$$= \bar{\hat{q}}_t(s_t, a_t^*(s_t, \alpha_t), \alpha_t) \geq \hat{q}_t(s_t, a_t^*(s_t, \alpha_t), \alpha_t) = q_t(s_t, a_t^*(s_t, \alpha_t), \alpha_t)$$

$$= \max_{a \in \mathcal{A}} q_t(s_t, a, \alpha_t) = \max_{a \in \mathcal{A}} \hat{q}_t(s_t, a, \alpha_t) = \hat{v}_t(s_t, \alpha_t),$$

where we first used the definition of $\bar{\pi}$, then used the definition of $\bar{\hat{q}}_T$, followed with the assumed property that $\bar{\hat{q}}_t$ is an upper envelope for $\hat{q}_t$. We then employ the equivalence of $\hat{q}_t$ and $q_t$ twice, the definition of $a_t^*$, and the relationship between $\hat{v}_t$ and $\hat{q}_t$ established in (30).

Next, we confirm that if $\bar{v}_t$ is an upper envelope for $\hat{v}_t$, then taking one step back and using the short-hand $\tilde{r}_{t'} = r(\tilde{s}_{t'}, \tilde{a}_{t'}, \tilde{s}_{t'+1})$ we get:

$$\bar{\hat{q}}_{t-1}(s_{t-1}, a_{t-1}, \alpha_{t-1}) :=$$

$$\text{VaR}_{\alpha_{t-1}}^{\tilde{a}_{t'} \sim \bar{\pi}_{t'}(\tilde{h}_{1:t'})} [r(s_{t-1}, a_{t-1}, \tilde{s}_t) + \sum_{t'=t}^{T} \tilde{r}_{t'} \mid s_{t-1}, a_{t-1}, \bar{\alpha}_{t-1}(\tilde{s}_{1:t-1}, \tilde{a}_{1:t-2}) = \alpha_{t-1}]$$

$$= \sup_{\zeta \in \Delta_S} \Big\{ \min_{s' \in \mathcal{S}} r(s_{t-1}, a_{t-1}, s') +$$

$$\text{VaR}_{\alpha_{t-1} \zeta_{s'} \boldsymbol{p}_{s'}^{-1}}^{\tilde{a}_{t'} \sim \bar{\pi}_{t'}(\tilde{h}_{1:t'})} [\sum_{t'=t}^{T} \tilde{r}_{t'} \mid \tilde{s}_t = s', a_{t-1}, \bar{\alpha}_{t-1}(\tilde{s}_{1:t-1}, \tilde{a}_{1:t-2}) = \alpha_{t-1}] \mid \alpha_{t-1} \cdot \boldsymbol{\zeta} \leq \boldsymbol{p} \Big\}$$

$$= \sup_{\zeta \in \Delta_S} \Big\{ \min_{s' \in \mathcal{S}} r(s_{t-1}, a_{t-1}, s') +$$

$$\text{VaR}_{\alpha_{t-1} \zeta_{s'} p_{s'}^{-1}}^{\tilde{a}_{t'} \sim \bar{\pi}_{t'}(\tilde{h}_{1:t'})} [\sum_{t'=t}^{T} \tilde{r}_{t'} \mid \tilde{s}_t = s', \bar{\alpha}_t(\tilde{s}_{1:t}, \tilde{a}_{1:t-1}) = g(\alpha_{t-1}, s_{t-1}, a_{t-1}, s')] \mid \alpha_{t-1} \cdot \boldsymbol{\zeta} \leq \boldsymbol{p} \Big\}$$

$$\geq \min_{s' \in \mathcal{S}} r(s_{t-1}, a_{t-1}, s') +$$

$$\text{VaR}_{g(\alpha_{t-1}, s_{t-1}, a_{t-1}, s')}^{\tilde{a}_{t'} \sim \bar{\pi}_{t'}(\tilde{h}_{1:t'})} [\sum_{t'=t}^{T} \tilde{r}_{t'} \mid s_t = s', \bar{\alpha}_t(\tilde{s}_{1:t}, \tilde{a}_{1:t-1}) = g(\alpha_{t-1}, s_{t-1}, a_{t-1}, s')]$$

$$= \min_{s' \in \mathcal{S}} r(s_{t-1}, a_{t-1}, s') + \bar{v}_t(s', g(\alpha_{t-1}, s_{t-1}, a_{t-1}, s'))$$

$$\geq \min_{s' \in \mathcal{S}} r(s_{t-1}, a_{t-1}, s') + \hat{v}_t(s', g(\alpha_{t-1}, s_{t-1}, a_{t-1}, s'))$$

$$= \min_{s' \in \mathcal{S}} r(s_{t-1}, a_{t-1}, s') + \max_{a \in \mathcal{A}} \hat{q}_t(s', a, g(\alpha_{t-1}, s_{t-1}, a_{t-1}, s'))$$

$$= \min_{s' \in \mathcal{S}} r(s_{t-1}, a_{t-1}, s') + \max_{a \in \mathcal{A}} q_t(s', a, g(\alpha_{t-1}, s_{t-1}, a_{t-1}, s'))$$

$$= \sup_{\zeta \in \Delta_S} \Big\{ \min_{s' \in \mathcal{S}} r(s_{t-1}, a_{t-1}, s') + \max_{a \in \mathcal{A}} q_t(s', a, \alpha_{t-1} \zeta_{s'} p_{s'}^{-1}) \mid \alpha_{t-1} \cdot \boldsymbol{\zeta} \leq \boldsymbol{p} \Big\}$$

$$= \sup_{\zeta \in \Delta_S} \left\{ \min_{s' \in \mathcal{S}} r(s_{t-1}, a_{t-1}, s') + \hat{v}_t(s', \alpha_{t-1}\zeta_{s'} p_{s'}^{-1}) \mid \alpha_{t-1} \cdot \zeta \leq \mathbf{p} \right\}$$

$$= \hat{q}_{t-1}(s_{t-1}, a_{t-1}, \alpha_{t-1}).$$

where

$$g(\alpha, s, a, s') := \alpha p_{s,a,s'}^{-1} \cdot \left[ \arg \max_{\zeta \in \Delta_S : \alpha \cdot \zeta \leq \mathbf{p}_{s,a}} \min_{s'' \in S} r(s, a, s'') + \max_{a \in \mathcal{A}} q_t(s, a, \alpha \zeta_{s''} p_{s,a,s''}^{-1}) \right]_{s'}$$

The first step comes from the decomposition of VaR, the second from the fact that $\bar{\alpha}_t$ is known given that $\bar{\alpha}_{t-1}$, $\tilde{s}_t$, and $\tilde{a}_{t-1}$ are fixed while $\sum_{t'=t}^{T} r(\tilde{s}_{t'}, \tilde{a}_{t'}, \tilde{s}_{t'+1})$ only depends on $\tilde{s}_t$ and $\bar{\alpha}_t$. The third step follows from replacing the supremum over $\zeta$ with a feasible member of $\Delta_S$. The fourth step follows from the definition of $\bar{v}_t$, followed with the fact that it upper bounds $\hat{v}_t$. Finally, we exploit the definition of $g(\cdot)$ and of $\hat{q}_{t-1}(\cdot)$. This completes the inductive proof demonstrating that $\hat{\bar{q}}_t(\cdot)$ is an upper envelope for $\hat{q}_t(\cdot)$ for all $1 \leq t \leq T$.

□

