}\,[z - \tilde{x}]_+\right) \; = \; \inf_{\boldsymbol{\xi} \in \Delta_m}\,\left\{\boldsymbol{\xi}^\top \boldsymbol{x} \mid \alpha \cdot \boldsymbol{\xi} \leq \boldsymbol{q}\right\}, \qquad (2)$$

with $\mathrm{CVaR}_0\,[\tilde{x}] = \mathrm{ess}\inf[\tilde{x}]$ and $\mathrm{CVaR}_1\,[\tilde{x}] = \mathbb{E}[\tilde{x}]$. Finally, the *entropic value at risk* (EVaR), with $\mathrm{EVaR}_0\,[\tilde{x}] = \mathrm{ess}\inf[\tilde{x}]$ and $\mathrm{EVaR}_1\,[\tilde{x}] = \mathbb{E}[\tilde{x}]$, is defined for $\alpha \in (0, 1]$ as (Ahmadi-Javid, 2012)

$$\mathrm{EVaR}_\alpha\,[\tilde{x}] = \sup_{\beta > 0}\frac{1}{\beta}\left(-\log \alpha^{-1}\mathbb{E}\big[\exp\left(-\beta\tilde{x}\right)\big]\right) = \inf_{\boldsymbol{\xi} \in \Delta_m : \boldsymbol{\xi} \ll \boldsymbol{q}}\left\{\boldsymbol{\xi}^T \boldsymbol{x} \mid \mathrm{KL}(\boldsymbol{\xi}\|\boldsymbol{q}) \leq -\log \alpha\right\}, \quad (3)$$

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

The direction $\Rightarrow$ follows from the definition of VaR (see Eq. 1), which implies that VaR upper-bounds any $z$ that satisfies the left-hand condition:

$$\mathbb{P}\left[\tilde{x} < z \mid \tilde{y} = i\right] \leq \zeta_i \Rightarrow \mathrm{VaR}_{\zeta_i}\left[\tilde{x} \mid \tilde{y} = i\right] = \sup\ \left\{z \in \mathbb{R} \mid \mathbb{P}\left[\tilde{x} < z \mid \tilde{y} = i\right] \leq \zeta_i\right\} \geq z.$$

In step (e), we solve for $z$. Finally, the form in (18) follows by replacing each $\zeta_i$ by $\alpha\zeta_i\hat{p}_i^{-1}$.   $\

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

The decomposition proposed in Proposition A.2 can now be used, exactly as was done for the case of VaR, to obtain a decomposition for the risk averse MDP:

$$
\max_{\pi \in \Pi} Q_\alpha^{\tilde{a} \sim \boldsymbol{\pi}(\tilde{s})}[r(\tilde{s}, \tilde{a}, \tilde{s}')] \; = \; \max_{\pi \in \Pi} \; \sup_{\boldsymbol{\zeta} \in [0,1]^S} \left\{ \min_{s \in \mathcal{S}: \zeta_s < 1} \; Q_{\zeta_s}^{\tilde{a} \sim \pi(s)}(r(s, \tilde{a}, \tilde{s}') \mid \tilde{s} = s) \mid \sum_{s=1}^{S} \zeta_s \hat{p}_s < \alpha \right\}
$$

$$
= \; \sup_{\boldsymbol{\zeta} \in [0,1]^S} \; \max_{\pi \in \Pi} \left\{ \min_{s \in \mathcal{S}: \zeta_s < 1} \; Q_{\zeta_s}^{\tilde{a} \sim \pi(s)}(r(s, \tilde{a}, \tilde{s}') \mid \tilde{s} = s) \mid \sum_{s=1}^{S} \zeta_s \hat{p}_s < \alpha \right\}
$$

$$
= \; \sup_{\boldsymbol{\zeta} \in [0,1]^S} \left\{ \min_{s \in \mathcal{S}: \zeta_s < 1} \left( \max_{\boldsymbol{d} \in \Delta_A} Q_{\zeta_s}^{\tilde{a} \sim \boldsymbol{d}}(r(s, \tilde{a}, \tilde{s}') \mid \tilde{s} = s) \mid \sum_{s=1}^{S} \zeta_s \hat{p}_s < \alpha \right) \right\}.
$$