# OpenReview forum: "On Dynamic Programming Decompositions of Static Risk Measures in Markov Decision Processes"
_NeurIPS.cc/2023/Conference — NeurIPS 2023 poster_

### Official Review · Reviewer_tkg9 · 2023-07-06

**Soundness:** 3 good
**Presentation:** 3 good
**Contribution:** 3 good
**Rating:** 6
**Confidence:** 4

**Summary:**

The paper shows that the dynamic programming decomposition methods used in risk-averse MDP are suboptimal for CVaR and EVaR, but optimal for VaR. These findings seem important as such decomposition methods become increasingly popular recently.

**Strengths:**

The findings that the decomposition fails for CVaR and EVaR are of importance and interesting to see.

**Weaknesses:**

The paper presents negative outcomes arising from the application of the decomposition method in CVaR and EVaR MDP. However, it does not offer a solution or rigorous analysis to address or evaluate these shortcomings. Particularly, in Theorem 3.2, the authors demonstrate that there exists a risk level where the equalities in (8) do not hold, indicating that the decomposition is suboptimal. While this finding is valuable, it raises crucial questions. For instance, under which parameter values does the optimality of the decomposition hold? If we randomly select a risk level from an interval, what are the chances of achieving optimal decomposition? Can we establish an upper bound on the loss incurred by using the decomposition? Is there an alternative form of decomposition that yields smaller losses? These questions hold significant relevance and would provide insights into handling the limitations.

Similar questions can be asked regarding the EVaR. Therefore, although the findings are noteworthy, the paper is not yet ready for publication. Further work on addressing the shortcomings of the decomposition method would significantly enhance the paper's quality.



**Questions:**

- Please refer to the questions asked above.
- Is there any way to numerically evaluate the suboptimality of the decomposition?

**Limitations:**

I do not see any negative societal impact from the paper.

---

> ### Author Rebuttal · Authors · 2023-08-09
>
> We thank the reviewer for the constructive comments.
>
> First, we need to clarify a fact that the reviewer might have missed while reading the paper. Namely, both CVaR and EVaR do possess valid dynamic programming-based solution methods, as presented in Bäuerle and Ott. (2011) and Hau et al. (2023) respectively. We, therefore, disagree with the claim that the paper "does not offer a solution" to the identified shortcomings. In other words, a solution is to rely on alternative decompositions, which have strong theoretical foundations. We realize that this message might not have been clear and plan on reinforcing it in the conclusion of the revised paper.
>
> For the case of VaR, we identified important technical issues with the analysis in Li et al. (2022) of its proposed dynamic programming decomposition. This is remedied in section 5 of our paper with a decomposition for the case $T=1$ that is rigorously supported. Based on the comments of reviewer bRij, we decided to include in the appendix of the revised version of the paper the implied theoretical result regarding the case of general $T>1$. This should provide a complete dynamic programming decomposition for the case of value-at-risk.
>
> Based on the discussion above, we therefore strongly disagree with the reviewer's claim that our "paper is not yet ready for publication". In its current form, our paper already demonstrates that a popular dual decomposition scheme that is used for CVaR and EVaR (and could continue to be deployed for other coherent risk functionals based on Pflug and Pichler (2016)) is suboptimal, thus encouraging the use of their alternate primal decomposition form, and rigorously identifies a decomposition scheme for VaR. Given the wide adoption of the problematic CVaR decomposition and the risk that reinforcement learning researchers or practitioners continue to incorrectly employ the decomposition presented in Pflug and Pichler (2016), this paper addresses important and timely objectives and successfully achieves them. Given that none of the reviewers has questioned the legitimacy of our mathematical claims, we can only conclude that it is indeed ready for publication.
>
> This being said, we appreciate the constructive feedback that was provided and, while we believe some of the questions fall outside the scope of our work, we are happy to provide some high-level answers.
>
> **Under which parameter values does the optimality of the decomposition hold?**
> The examples in Theorems 3.2 and 4.1 can be readily adapted to any risk level in $\alpha \in (0,1)$. In fact, the decompositions proposed by Chow et al. (2015) are only guaranteed to be optimal for $\alpha = 0$ or $\alpha = 1$, where the objective simply reduces to the expected value and the worst case, respectively. In contrast, the decomposition in Bäuerle and Ott. (2011) is optimal for all $\alpha\in[0,1]$.
>
> **What are the chances of achieving optimal decomposition?**
> It is not clear to us under what distribution the reviewer is interested in measuring a probability of optimal decomposition. One could, in fact, easily create a population of counterexamples for which the CVaR and EVaR decomposition would be sub-optimal with probability one. On the other hand, Min et al. (2022) claim to have identified a family of execution problems (a form of portfolio optimization) where the decomposition is exact (with probability one) due to the convexity of the objective with respect to the deterministic policy.
>
> **Can we establish an upper bound on the loss incurred by using the decomposition?**
> Using a simple scaling argument (on the rewards), the example in Theorem 3.2 shows that the error in worst-case risk estimation can be arbitrarily large, even when $T = 1$ Perhaps, one could establish an upper bound on the suboptimality error under a specific MDP structure, but it is not clear that this would be useful since the decomposition in Bäuerle and Ott. (2011) is exact. Please also refer to the example (Figure 2) in the attached pdf submitted along with the global response.
>
> **Is there an alternative form of decomposition that yields smaller losses?**
> Yes, see Bäuerle and Ott. (2011) and Hau et al. (2023) for CVaR and EVaR, respectively.
>
> **Is there any way to numerically evaluate the suboptimality of the decomposition?**
> Yes, by implementing the primal decomposition scheme presented in Bäuerle and Ott. (2011) and Hau et al. (2023). Note that in the latter, the authors provide a discretization scheme (for the continuous state variable) for which the estimation error can be bounded.
>
> **References**
>
> [1] Nicole Bäuerle and Jonathan Ott. Markov decision processes with average-value-at-risk criteria. Mathematical Methods of Operations Research, 74(3):361–379, 2011.
>
> [2] Yinlam Chow, Aviv Tamar, Shie Mannor, and Marco Pavone. Risk-sensitive and robust decision-making: A CVaR optimization approach. In Neural Information Processing Systems (NIPS),2015.
>
> [3] Jia Lin Hau, Marek Petrik, and Mohammad Ghavamzadeh. Entropic risk optimization in discounted MDPs. In Artificial Intelligence and Statistics (AISTATS), 2023.
>
> [4] Xiaocheng Li, Huaiyang Zhong, and Margaret L. Brandeau. Quantile Markov decision processes. Operations Research, 70(3):1428–1447, 2022.
>
> [5] Seungki Min, Ciamac C Moallemi, and Costis Maglaras. Risk-sensitive optimal execution via a conditional value-at-risk objective. arXiv preprint arXiv:2201.11962, 2022.
>
> [6] Georg Ch Pflug and Alois Pichler. Time-consistent decisions and temporal decomposition of coherent risk functionals. Mathematics of Operations Research, 41(2):682–699, 2016.

---

> > ### Comment · Reviewer_tkg9 · 2023-08-15
> >
> > I thanks the authors for the responses. I think they have provided a strong rebuttal and seem to convince me that their contributions are significant and ready for publication. I will raise my score.

---

### Official Review · Reviewer_eWnC · 2023-07-06

**Soundness:** 3 good
**Presentation:** 2 fair
**Contribution:** 3 good
**Rating:** 6
**Confidence:** 1

**Summary:**


The paper tackles the risk-level decomposition of risk measures in the finite Markov Decision Process (MDP). The work firstly points out the suboptimality of the decomposition proposed in prior works (Conditional-Value-at-Risk), by mathematically proving that the decomposition in CVaR does not maintain the equality to the optimal policy under policy optimization.

Then the work mathematically proves the decomposition in Entropy-Value-at-Risk (EVaR) fails to find the minimum for the policy evaluation. Following that, the paper corrects the decomposition of EVaR and proposes a new dynamic program decomposition for Value-at-Risk (VaR). By avoiding the saddle-point gap in policy optimization, the new method remains optimal for both policy optimization and policy evaluation.


**Strengths:**

The method contributes to identifying the suboptimality in policy optimization and policy evaluation in prior methods (CVaR and EVaR). The paper points out the reason causing the suboptimality and proposes a new method for improvement. As the paper improves the correctness of the policy learning in risk-averse methods, the theoretical result can also be significant when people apply the risk-averse approach in practice.

The figures are clear and help with understanding the counterexamples.


**Weaknesses:**

I understand that the paper mainly focuses on theoretical analysis, and it makes sense to me that there is no empirical evidence for the correctness of the new method. One small suggestion is to add a numerical experiment.

**Questions:**

N/A

**Limitations:**

Yes

---

> ### Author Rebuttal · Authors · 2023-08-09
>
> Thank you for your review and suggestions. Adding a numerical experiment is a good suggestion. In the final version of the paper, we will add a simple example that demonstrates how the duality gap translates into a sub-optimal policy. Please see the attached pdf with the global response, where we describe a small example that demonstrates numerically the gap in a simple MDP.

---

> > ### Comment · Reviewer_eWnC · 2023-08-18
> > **Reply**
> >
> > I appreciate the authors for the reply. I have read it and will maintain the score.

---

### Official Review · Reviewer_Gp5S · 2023-07-06

**Soundness:** 4 excellent
**Presentation:** 4 excellent
**Contribution:** 3 good
**Rating:** 7
**Confidence:** 3

**Summary:**

The main contribution of the paper is to demonstrate that a popular dynamic programming decomposition for well-known risk metrics only provides suboptimal policies. More precisely, the underlying idea of such decompositions is to operate on an extended state space that incorporates a continuous state variable between 0 and 1 capturing the system risk level at the current stage. The belief in these existing decompositions was that the risk state variable could be finely discretized to approximate the optimal value function arbitrarily. However, the authors show a counter-example for CVaR and EVaR. Finally, they show that VaR does admit such a decomposition with a new proof scheme.

**Strengths:**

+ The paper and proofs are very well-written and structured
+ Contribution is impactful within the analysis of risk measures

I particularly find that this is an outstanding paper and contribution. First, it closes an open theoretical question on a meaningful and growing body of literature composed of 10+ papers only in these last five years, specifically demonstrating via a relatively simple but ingenious example that discretization is necessarily sub-optimal. The results are intriguing because the discretization assumption was a rather intuitive result, justifying many of the attempted proofs in the most recent literature. The authors also provide corrected decompositions for EVaR and VaR.

I also note that it is also a bit concerning that these results contradict the optimality of many recent works, which are thus worth revisiting in the future. (On that note, I verified the proofs and examples of this paper to the best of my abilities, but did not re-analyze the previous papers' proofs, except to re-check their statements).



**Weaknesses:**

- Much of the intuition is hidden in the proofs.
- Lack of discussion of possible research avenues to address this issue.

My primary concern is that the presentation can be excessively technical in some parts. I believe the proofs are very well written and easy to follow, especially as they are mostly counter-examples, but it would be interesting if authors could bring to light what fails more intuitively (just the saddle point discussion was commented on briefly). In particular, my understanding from the examples is that the discretization violates a certain variant of Von Neummann's minimax theorem due to non-convexity/non-concavity of policy choice, here captured in the strict inequality of equation (10). Perhaps my understanding is not accurate, but nonetheless it would be interesting if authors could expand more on the saddle point argument, perhaps providing directions on conditions for other coherent metrics that do not suffer from the same discretization issue.


Minor notes:

- Please kindly use capital letters when naming lemmas and theorems, e.g., Lemma X as opposed to lemma X.
- In the proof of Prop. 3.1, "Define" should be "define"; also please define "ess inf".
- In the proof of Theorem 4.2, I am a bit confused with line l.457: "the first inequality follows from adding a constraint on the pairs."



**Questions:**

1. The next natural question is how poor the discretized systems can be. Are there any asymptotic guarantees on its quality, even under some non-zero approximation ratio?
2. Are there specific coherent metrics where the discretization could still be valid?


**Limitations:**

The authors have not included any limitation subsection.

---

> ### Author Rebuttal · Authors · 2023-08-09
>
> Thank you for the detailed review and the encouraging words. We answer the questions and clarify some of the points below.
>
> **Lack of discussion of possible research avenues to address this issue.**
> This is a great point. Given how widespread this error is in the extant literature, we believe that it is important to correct this important misunderstanding. As we discuss in the introduction, the alternative algorithms proposed in Bauerle \& Ott (2011) and Hau et al. (2023) do not suffer from the same issue. We will clarify the relevant discussion in the introduction and throughout the paper in order to emphasize this issue.
>
> **The next natural question is how poor the discretized systems can be. Are there any asymptotic guarantees on its quality, even under some non-zero approximation ratio?**
> That is an intriguing question. We attach the sub-optimality plot of the CVaR of the return for this simple MDP in the attached pdf with the global response, our examples indicate that the error can be arbitrarily large without additional structural assumptions on the MDP. One could also refer to the $\theta$ function, as shown in Figure 2 of our paper, and manually design a MDP to make CVaR and EVaR algorithms incur arbitrary loss.
>
> **Are there specific coherent metrics where the discretization could still be valid?**
> That is a good question, but we are unaware of ***coherent*** risk measures for which this type of decomposition would work. Given that the robust representation of all coherent risk measures involves a minimization, the same problem is likely to be present.  Note that the decomposition works for VaR, but that is not a coherent risk measure.
> The decomposition of Bauerle \& Ott (2011) and Hau et al. (2023) for CVaR and EVaR, respectively, is done in the primal form of the risk measures and therefore does not suffer from the duality gap which we present in this paper.
>
> **Providing more intuition about the results**
> In the revision, we will make sure to include additional discussion regarding the saddle point condition that is violated by faulty decompositions for CVaR and EVaR. In particular, in the context of convex (thus continuous) action space, it might be possible for some applications to exploit the concavity of the long-term reward function to establish the needed saddle point property. One example already appears in Min et al. (2022) under the shape of a portfolio management problem.
>
> **References**
>
> [1] Nicole Bäuerle and Jonathan Ott. Markov decision processes with average-value-at-risk criteria. Mathematical Methods of Operations Research, 74(3):361–379, 2011.
>
> [2] Jia Lin Hau, Marek Petrik, and Mohammad Ghavamzadeh. Entropic risk optimization in discounted MDPs. In Artificial Intelligence and Statistics (AISTATS), 2023.
>
> [3] Seungki Min, Ciamac C Moallemi, and Costis Maglaras. Risk-sensitive optimal execution via a conditional value-at-risk objective. arXiv preprint arXiv:2201.11962, 2022.

---

> > ### Comment · Reviewer_Gp5S · 2023-08-13
> >
> > Thank you for the careful and detailed responses to my questions. I believe the addition of the case T > 1 mentioned to the other reviewers would certainly better highlight the contribution of the work.

---

### Official Review · Reviewer_bRij · 2023-07-11

**Soundness:** 2 fair
**Presentation:** 2 fair
**Contribution:** 2 fair
**Rating:** 3
**Confidence:** 3

**Summary:**

The authors demonstrate that dynamic programming decompositions for CVaR and EVaR can be suboptimal, and show that VaR may not be subject to the same problems in MDPs of horizon 1.

**Strengths:**

The paper provides some interesting observations conflicting with previous papers on CVaR and EVaR optimization in RL.

**Weaknesses:**

While this paper makes some interesting observations about the decompositions of CVaR, EVaR, and VaR, it seems like an incomplete paper, without enough contributions meriting acceptance.

The “dynamic program” presented in Section 5 seems far from implementable, with weak (if any) guarantees. For contrast, [Chow 2015] shows that the CVaR Bellman operator is a contraction with a unique fixed point, and also discuss how to discretize the space of $\alpha$ to make practical implementations possible. They also provide proof-of-concept experiments. The proposed VaR operator seems to face similar issues (e.g., with $\alpha$ being continuous), but the submitted paper appears to handle none of these points.  It would be interesting to see the decomposition proposed in Theorem 5.2 translated into a policy learning algorithm, e.g., of the style of value iteration proposed in [Chow 2015], complete with theoretical analysis and empirical evaluation.

Additionally, Theorems 5.1 and Proposition 5.3 appear to only hold for $T=1$ which is essentially a contextual bandit? While the authors state that results can be extended to $T > 1$ using “standard techniques,” I find that $T = 1$ is an unusual assumption in RL (for MDPs) and usually something that makes risk-sensitive RL much easier (because you don’t have to deal with stochasticity from transitions). In this light, it seems worth it to at least expand Section 5 for $T > 1$.

**Questions:**

- In policy evaluation, will the proposed algorithm need to be able to handle continuous $\alpha$, and how can that be done?

- Is it possible to learn a policy with $T > 1$, and what would the specific policy learning algorithm look like (in broad strokes)?

**Limitations:**

yes

---

> ### Author Rebuttal · Authors · 2023-08-09
>
> We thank the reviewer for the constructive comments, which encourage us to suggest adding to the appendix of the paper a natural extension of the VaR decomposition presented as Theorem 5.2 to the general case of $T>1$ (more on this below). Indeed, we had not noticed that a reader might consider Section 5 incomplete without a proper derivation of a dynamic program decomposition method for $T>1$, and we are happy to remediate this issue.
>
> This being said, we strongly disagree with the idea that our paper might be "incomplete". Indeed, the goal of our paper is to demonstrate that a popular dual decomposition scheme that is used for CVaR and EVaR (and could continue to be deployed for other coherent risk functionals based on Pflug and Pichler (2016)) is suboptimal, thus encouraging the use of their alternate primal decomposition form, and rigorously identifies a DP decomposition scheme for VaR. Given the wide adoption of the problematic CVaR decomposition and the risk that reinforcement learning researchers or practitioners continue to incorrectly employ the decomposition presented in Pflug and Pichler (2016), this paper addresses important and timely objectives and successfully achieves them.
>
> We also disagree with the idea that our paper focuses on contextual bandit problems. The decision to identify counterexamples that focus on an MDP with $T=1$ is done without loss of generality and intentionally with the aim of simplifying the technical exposition. We, however, realized when reading the comment that the decomposition in Theorem 5.3 was perhaps overly simplified and are happy to include in an appendix the more general version of this theorem.
>
> **Theorem**. Given any finite MDP with horizon $T>0$ and $\alpha\in[0,1]$, we have
>
> $\max\_{\pi\in\Pi} VaR\_\alpha^{\tilde a_t\sim\pi_t(\tilde s_t)}[\sum_{t=1}^T r(\tilde s_t, \tilde a_t, \tilde s_{t+1})]$
> $= \sup_{\zeta\in \Delta(S)} (  \min_{s\in\mathcal{S}} \max_{a\in A } Q_1(s,a,\alpha \zeta_s \hat p_s^{-1})  |  \alpha\cdot \zeta \leq \hat p )$.
>
> with $Q_{T+1}(s,a,\alpha) := 0$ if $\alpha<1$ otherwise infinity, and for all $1\leq t \leq T$:
>
> $Q_{t}(s,a,\alpha) :=  \sup_{\zeta\in\Delta(S)} (  \min_{s'\in\mathcal{S}} r(s,a,s')+\max_{a'\in A } Q_{t+1}(s',a',\alpha \zeta_s p_{sas'}^{-1}) |  \alpha\cdot \zeta \leq p_{sa})$
>
> Moreover, the policy
>
> $\bar \pi_t(s_{1:t},a_{1:t-1})\in \arg\max_{a\in A} Q_t(s_t,a,\bar \alpha_t(s_{1:t},a_{1:t-1}))$
>
> with $\bar \alpha_0:=\alpha$ and
>
> $\bar \alpha_t(s_{1:t},a_{1:t-1}) \in  \bar \alpha_{t-1}(s_{1:t-1},a_{1:t-2}) p_{s_{t-1},a_{t-1},s_t}^{-1}\cdot$
>
> $[\arg\max_{\zeta\in\Delta(S):\bar \alpha_{t-1}\cdot\zeta\leq p_{s_{t-1},a_{t-1}}}\min_{s\in S} \max_{a\in A} Q_t(s,a,\bar \alpha_{t-1}(s_{1:t-1},a_{1:t-2})\zeta_s p_{s_{t-1},a_{t-1},s_t}^{-1})]_{s_t}$
>
> is optimal.
>
> While the proof is inspired by the work of Li et al. (2022), it corrects for the wrong definition of VaR that is used, circumvents an issue in the proof of Lemma 2, extends the results to the space of stochastic policies, and rigorously confirms that the supremum over $\zeta\in\Delta(S)$ is achieved in order to employ it in the definition of the optimal policy.
>
> Regarding the implementability of the decomposition for VaR, we first recall that the objective of this paper is NOT to propose a new algorithm for VaR-MDP but rather to solidify/question the foundation of methods that have been recently proposed and popularized by proceedings/journals with the highest standards in terms of quality. If the reviewer is interested in implementation details and empirical evaluation, we encourage him/her to refer Li et al. (2022), where section 4 proposes an algorithm for the VaR MDPs that is then deployed in an HIV treatment initiation problem (see section 6.2). We believe this should answer most of the reviewer's questions and perhaps convince him/her that the scope of our paper is justified.
>
> Regarding the existence of a unique fixed point, please note that such a question does not apply to the finite time horizon setting that we consider. We agree, however, that it is a relevant question for extending the decomposition to an infinite horizon setting, but we judged this to be beyond the scope of this paper, given its objectives. There has been relevant work studying this question in the field of distributional RL and target-based Markov decision processes, but this is a complex topic that deserves careful treatment.
>
> **References**
>
> [1] Xiaocheng Li, Huaiyang Zhong, and Margaret L. Brandeau. Quantile Markov decision processes. Operations Research, 70(3):1428–1447, 2022.
>
> [2] Georg Ch Pflug and Alois Pichler. Time-consistent decisions and temporal decomposition of coherent risk functionals. Mathematics of Operations Research, 41(2):682–699, 2016.

---

### Official Review · Reviewer_Vzn6 · 2023-07-20

**Soundness:** 4 excellent
**Presentation:** 3 good
**Contribution:** 4 excellent
**Rating:** 7
**Confidence:** 3

**Summary:**

This paper investigates the effectiveness of decomposition approaches for solving risk-averse Markov Decision Processes (MDPs) with Conditional Value-at-Risk (CVaR), Expected Value-at-Risk (EVaR), and Value-at-Risk (VaR) objectives. The goal is to assess the validity and accuracy of these common decomposition techniques for risk-averse decision-making.

Main contributions:

Sub-optimality of CVaR and EVaR decomposition: The paper demonstrates that the widely used decomposition approaches for CVaR and EVaR objectives are inherently suboptimal, invalidating previous works. These methods involve a saddle-point gap during policy optimization, leading to incorrect results. As a result, practitioners should exercise caution when using these decomposition techniques for risk-averse MDPs.

Optimal VaR decomposition: In contrast to the suboptimal CVaR and EVaR decompositions, the paper identifies the VaR decomposition as an optimal approach for both policy evaluation and optimization in risk-averse MDPs. The VaR decomposition does not suffer from the same saddle-point problem, making it a reliable method for risk-averse decision-making.

Increased awareness and call for alternative approaches: The findings highlight the limitations of two traditional decomposition methods, CVaR and EVaR, and emphasize the need for scrutiny when using these decompositions. Researchers are encouraged to explore alternative approaches, such as parametric dynamic programs, to improve risk-averse decision-making in MDPs.

**Strengths:**

Rigorous analysis: The main strength of the paper is its rigorous treatment of three decomposition approaches for risk-averse MDPs.

Significant and clearly stated contribution for the field: The paper shows the sub-optimality of CVaR and EVaR decomposition methods invalidating several previous works on the topic. This finding is valuable as it warns practitioners about unknown limitations of these techniques in risk-averse decision-making.

Optimal VaR decomposition: By identifying the VaR decomposition as an optimal approach for policy evaluation and optimization in risk-averse MDPs, the paper provides a practical solution to the limitations of CVaR and EVaR. Thus this VaR decomposition is a reliable alternative to these two methods.

Encouraging further research: In light of the limitations faced by CVaR and EVaR, the paper calls for exploring alternative approaches, such as parametric dynamic programs.

Clarity in presentation and methodology: The paper is well-written. It uses clear explanations, comprehensive mathematical analysis, and compelling illustrative examples.

**Weaknesses:**

Limited set of approaches: Perhaps the main weakness of the paper is the limited set of (three) decomposition approaches being scrutinized. While the paper highlights the sub-optimality of CVaR and Expected Value-at-Risk EVaR methods, it does not compare a broader range of state-of-the-art techniques. This limitation makes unclear the extent to which VaR decomp Google Mapsosition can be considered optimal. I appreciate that it would not be possible, given length constraints and scope of the paper, to evaluate other baselines comprehensively. However, it would be valuable if the authors could mention other state-of-the-art techniques in a related works section, and whether they can say anything about them in relation to the approaches they investigated here. For instance, control as inference and active inference use the KL divergence as a decision-making objective (i.e., a belief-based reward function) which entails risk-averse behaviour, and which might be related to VaR [1,2,3].

Unclear scalability discussion: The paper does not extensively discuss the scalability of the proposed VaR decomposition concerning problem size or dimensionality. I appreciate that this might not be important for theoretically minded readers, however, since the paper has important implications for practitioners (i.e., widespread adoption of VaR) it would be nice to immediately know, via some discussion or suitable reference, how well this method scales as MDPs grow complex and large in applications.

Limited generalization: The paper comprehensively addresses the setting of risk-averse MDPs. It would be nice to know whether the theoretical analyses presented here can offer some insights on more general problem classes like risk averse partially observable Markov decision processes (POMDPs). For example, has VaR been extended to risk averse POMDPs, and if so would you expected it to be optimal in this case?

From the two points above, one of the main weaknesses of this paper is that it is not straightforward to infer to what extent VaR is a viable approach in a wide range of practical problems, and thus it is unclear to what extent the paper implies an optimal, promising in the near long-term approach to the risk-averse decision problem. In my understanding, this is the main point of broader impact in the machine learning community, so it should be addressed.

Typos: l 136, l200

[1]	S. Levine, ‘Reinforcement Learning and Control as Probabilistic Inference: Tutorial and Review’, arXiv:1805.00909 [cs, stat], May 2018, Accessed: Dec. 29, 2021. [Online]. Available: http://arxiv.org/abs/1805.00909
[2]	L. Da Costa, N. Sajid, T. Parr, K. Friston, and R. Smith, ‘Reward Maximization Through Discrete Active Inference’, Neural Computation, vol. 35, no. 5, pp. 807–852, Apr. 2023, doi: 10.1162/neco_a_01574.
[3]	D. Hafner, P. A. Ortega, J. Ba, T. Parr, K. Friston, and N. Heess, ‘Action and Perception as Divergence Minimization’, arXiv:2009.01791 [cs, math, stat], Oct. 2020, Accessed: Nov. 07, 2020. [Online]. Available: http://arxiv.org/abs/2009.01791

**Questions:**

Can the authors elaborate on their selection of the specific decomposition approaches used for comparison? Are there any reasons why these three approaches were chosen over others?

Could the authors provide more insights into the computational complexity and scalability of the VaR decomposition method concerning the size and dimensionality of the risk-averse MDPs, either in a short discussion or references? Are there any notable challenges when applying this approach to large and more complex decision problems?

As a broader point of interest, how significant is the choice of the risk level (alpha) in the performance and efficiency of the VaR decomposition method? Are there any guidelines on selecting appropriate alpha values based on a given problem characteristics? This is important to understand the robustness of this method in a new risky environment, a desideratum for any risk averse decision-making algorithm in high stakes applications, which relates to the broader impact of this work as hinted in the last point of the conclusion.

Given that the paper focuses on risk-averse MDPs, are there any indications or early results on how the VaR decomposition approach and its optimality might generalize to other types of decision-making frameworks, such as partially observed problems?


**Limitations:**

The paper does a good job at addressing its limitations insofar as it focuses on three specific approaches to risk averse decision making.

The main limitation which has not been addressed, and which in my opinion should be addressed, is giving more context as to why this three approaches were chosen, or why is it is sensible to choose them, and mention the fact that there exist other state-of-the-art approaches, which need to be considered and comprehensively evaluated in the future, and which could serve as avenues for future research in addition to parametric dynamic programs, e.g., control as inference and active inference.

---

> ### Author Rebuttal · Authors · 2023-08-09
>
> We appreciate the detailed review and the helpful suggestions. We provide some clarification and answer your questions below.
>
> **It would be valuable if the authors could mention other state-of-the-art techniques in a related works section.**
> That is a good suggestion; thank you. In fact, the introduction describes multiple alternative state-of-the-art techniques that do not suffer from the problems we identified. We will edit the final version to emphasize the existence of these alternative approaches in the introduction and throughout the paper.
>
> **Can the authors elaborate on their selection of the specific decomposition approaches used for comparison? Are there any reasons why these three approaches were chosen over others?**
> These are good questions. We chose these decompositions because they are popular in modern literature and, as we demonstrate, do not offer the optimality guarantees that the papers claims. Also, these decomposition algorithms use a technique pioneered by Pflug and Pichler (2016).
>
> As we mention above, other decomposition techniques are worth further study. We hope our results will encourage further research into these alternative approaches, which do not suffer from the same issues we identify in this work. In the final version, we will position these approaches in the context of the three papers proposed by the reviewer.
>
> **Unclear the extent to which VaR decomposition can be considered optimal. How well this method scales as MDPs grow complex and large in applications.**
> This is a good point. Using the decomposition in large-scale MDPs, one will need to introduce further approximations. This complex issue is beyond this paper's scope and deserves systematic study in future work. The techniques proposed in section 4.2 of Li et al. (2022) and Chow et al. (2015) can be used with VaR decomposition too.
>
> **Could the authors provide more insights into the computational complexity?**
> While we agree with the reviewer that this is an important practical question to ask oneself, this paper focuses on the theoretical soundness of the decomposition schemes for VaR, CVaR, and EVaR. We will edit the narrative to further emphasize that the implementation concerns can be found in section 4.3 of Li et al. (2022) and that we leave the question of the optimal algorithm (in terms of complexity) open for future research.
>
> **How significant is the choice of the risk level?**
> Although this is an important concern, we must defer it to future work for similar reasons as stated above.
>
> **How the VaR decomposition approach... might generalize to other types of decision-making frameworks, such as partially observed problems?**
> We agree that this is an important line of work but is beyond this paper's scope. However, one could apply the quantile MDP decomposition scheme to POMDPs after translating them to the equivalent belief-space MDP. Of course, many technical details need to be addressed carefully in doing so.
>
> **This is the main point of broader impact in the machine learning community.**
> We actually expect our work to have a broad impact on ML, particularly by retarding the propagation of incorrect claims in the risk-averse RL literature. While we are convinced that the quantile MDP decomposition has the potential to be of great help in practice, our paper's main objectives when defining this decomposition are two-fold: 1) correct some technical inaccuracies in Li et al. (2022), and 2) provide insights on conditions under which optimal dynamic programming can be used for policy optimization.
>
> **References**
>
> [1] Yinlam Chow, Aviv Tamar, Shie Mannor, and Marco Pavone. Risk-sensitive and robust decision-making: A CVaR optimization approach. In Neural Information Processing Systems (NIPS),2015.
>
> [2] Xiaocheng Li, Huaiyang Zhong, and Margaret L. Brandeau. Quantile Markov decision processes. Operations Research, 70(3):1428–1447, 2022.
>
> [3] Georg Ch Pflug and Alois Pichler. Time-consistent decisions and temporal decomposition of coherent risk functionals. Mathematics of Operations Research, 41(2):682–699, 2016.

---

> > ### Comment · Reviewer_Vzn6 · 2023-08-15
> > **Thank you for these helpful clarifications and revisions**
> >
> > Thank you for your clarifications and intended revisions. The answers to my questions are helpful, and I believe that your intended revisions will strengthen the paper, help clarifying the paper's contribution and its broader impact, and make it more appealing to a broader audience.

---

### Author Rebuttal · Authors · 2023-08-09

We would like to take this opportunity to provide a holistic response to the reviewers' main questions and concerns below.

**Concern 1: Paper does not present a complete algorithm or numerical results**

This paper aims to correct the literature on risk-averse RL and retard the propagation of incorrect claims. As we discuss in detail in the introduction, numerous recent and well-cited papers in risk-averse RL are based on a fundamentally flawed technique; consequently, the central theorems of these papers are fundamentally incorrect. These fundamental errors that have been repeated in multiple papers are due to the fact that they devote only cursory attention to technical issues, which appear minor, but in the end, invalidate the approach.

To achieve the paper's main goal, we decided to focus primarily on the errors in prior work for the following reasons. First, focusing on implementation concerns would limit our ability to discuss the technical flaws of prior work in sufficient detail. It would divert one's attention from the technical flaws that we are identifying. Second, the implementation concerns that we omit have been discussed widely in prior work. Finally, as we point out in the paper, we believe that the CVaR and EVaR methods we study here are fundamentally flawed, and our goal is to steer the community to more promising techniques based on the decompositions studied in Bauerle \& Ott (2011) and Hau et al. (2023).

We will make three improvements to address some of the concerns raised in the review. First, we will clarify some of the writing to more precisely point the interested reader where to find the implementation details. Second, we will include the general multi-stage version ($T > 1$) of the VaR Bellman function, which corrects for the incorrect definition of VaR in Li et al. (2022) and confirms that the supremum is both attained and deterministic in the optimal policy.

**Theorem**. Given any finite MDP with horizon $T>0$ and $\alpha\in[0,1]$, we have

$\max\_{\pi\in\Pi} \operatorname{VaR}\_\alpha^{\tilde a_t\sim\pi_t(\tilde s_t)}[\sum_{t=1}^T r(\tilde s_t, \tilde a_t, \tilde s_{t+1})]= \sup\_{\zeta\in \Delta(S)} (  \min_{s\in\mathcal{S}} \max_{a\in A } q_1(s,a,\alpha \zeta_s \hat p_s^{-1})  |  \alpha\cdot \zeta \leq \hat p )$.

with $q_{T+1}(s,a,\alpha) := 0$ if $\alpha<1$ otherwise infinity, and for all $1\leq t \leq T$:

$q_{t}(s,a,\alpha) :=  \sup_{\zeta\in\Delta(S)} (  \min_{s'\in\mathcal{S}} r(s,a,s')+\max_{a'\in A } q_{t+1}(s',a',\alpha \zeta_s p_{sas'}^{-1}) |  \alpha\cdot \zeta \leq p_{sa})$

Third, we will include a numerical example demonstrating the suboptimality of the decomposition. Please see the attached pdf for the diagram of the MDP and the plot of the CVaR returns of the optimal policy and the policy computed using the decomposition in Chow et al. (2015).

**Question/Concern 2: Is it possible to quantify the suboptimality of the decompositions for CVaR and EVaR?**

Our counterexamples indicate there is no immediate additive or multiplicative bound on the suboptimality in the CVaR and EVaR decompositions. This is analogous to the duality gap for non-convex optimization, which can be arbitrarily large, depending on the structure of the optimization problem. While the suboptimality gap of the methods may vanish for MDPs with some specific structure, we are not convinced that this question is worth pursuing at this point. We advocate in the paper for the use of algorithms proposed by Bauerle \& Ott (2011) and Hau et al. (2023), which do not suffer from the optimality gap issues and are conceptually and computationally simpler.

We concur with the reviewers that a numerical study of the decompositions would be valuable. However, given the number of implementation and domain choices one needs to make and analyze in the implementation of these algorithms, we concluded that a cursory numerical study would not add much value to this theoretical paper. The numerical performance of the risk-averse RL algorithms is driven by a large number of implementation decisions, such as how the augmented state space is discretized, what method is used to compute the decomposed Bellman update, which value function approximation technique is used (quantile regression or least squares on CDF).

Even though an extensive numerical is beyond the scope of this work, we hope that a small illustrative example can alleviate some of the reviewers' concerns. In the final version of the paper, we will add a simple example that demonstrates how the duality gap translates into a suboptimality of a policy. We attach the plot of the CVaR of the return for this simple MDP in the attached pdf.

**References**

[1] Nicole Bäuerle and Jonathan Ott. Markov decision processes with average-value-at-risk criteria. Mathematical Methods of Operations Research, 74(3):361–379, 2011.

[2] Yinlam Chow, Aviv Tamar, Shie Mannor, and Marco Pavone. Risk-sensitive and robust decision-making: A CVaR optimization approach. In Neural Information Processing Systems (NIPS), 2015.

[3] Jia Lin Hau, Marek Petrik, and Mohammad Ghavamzadeh. Entropic risk optimization in discounted MDPs. In Artificial Intelligence and Statistics (AISTATS), 2023.

[4] Xiaocheng Li, Huaiyang Zhong, and Margaret L. Brandeau. Quantile Markov decision processes. Operations Research, 70(3):1428–1447, 2022.

---

### Decision · Program_Chairs · 2023-09-21

**Decision:**

Accept (poster)

**Comment:**

This paper deals with the sub-optimality of Dynamic Programming decomposition for risk measures like CVar and EVar.
The paper provides an interesting and significant contribution and is well-written and sound.
On the other hand, the reviewers raised some limitations about the limited number of risk measures considered and the lack of discussion about the scalability of the proposed decompositions.
The authors' rebuttals effectively solved most of the reviewers' issues, and even if they did not reach a consensus, there was a large support for accepting this paper.
The authors need to consider the reviewers' suggestions while preparing the final version of their paper.